# Efficient Sequence Packing without Cross-contamination: Accelerating Large Language Models without Impacting Performance

## Abstract

Effective training of today's large language models (LLMs) depends on large batches and long sequences for throughput and accuracy. To handle variable-length sequences on hardware accelerators, it is common practice to introduce padding tokens, so that all sequences in a batch have the same length. We show in this paper that the variation in sequence lengths in common NLP datasets is such that up to 50% of all tokens can be padding. In less common, but not extreme, cases (e.g. GLUE-cola with sequence length 128), the ratio is up to 89%. Existing methods to address the resulting inefficiency are complicated by the need to avoid 'cross-contamination' in self-attention, by a reduction in accuracy when sequence ordering information is lost, or by customized kernel implementations only valid for specific accelerators. This paper introduces a new formalization of sequence packing in the context of the well-studied bin packing problem, and presents new algorithms based on this formulation which, for example, confer a 2x speedup for phase 2 pre-training in BERT. We show how existing models can be adapted to ensure mathematical equivalence between the original and packed models, meaning that packed models can be trained with existing pre-training and fine-tuning practices.

## 1  Introduction

Many language datasets, including the de-facto pre-training dataset for BERT—Wikipedia, have a skewed distribution of sequence lengths (see Figure 1). However, typical machine learning accelerators, and their corresponding libraries, exhibit poor performance when processing variable-length workloads. A simple mitigation is to set a maximum sequence length, and to pad shorter sequences with padding tokens. This naive batching is widely used and provided in the vanilla BERT implementation as well as the Hugging Face framework [32]. Its effect is enhanced by the offline dataset generation process which, in BERT, attempts to "pack" together sentences so as to fill the sequence length as completely as possible [8]. We improve this process at a whole-dataset level.

We show that, even after this pre-processing, padding tokens represent $50\%$ of all tokens of the Wikipedia pre-training dataset at sequence length 512. Thus, by avoiding processing the padding tokens one can get a 2x speed-up for phase 2. Overall, the lengths range between $5$ tokens up to $512$. Samples of length 512 represent only $23.5\%$ of the dataset,

Beyond the simple batching, other solutions have been addressed in the literature, and in open-source software implementations. When processing sequences, most libraries and algorithms mention packing as reference to concatenating sentences from the same document (BERT) or from different documents (BERT, T5 [24], GPT-3 [4], and RoBERTa [16]) as they arrive (GREEDY) from the source dataset to generate the training dataset. None of the respective papers addresses the packing

efficiency, i.e., remaining fraction of padding. To "separate" sequences from different documents, a separator token is introduced. However, this is not sufficient and can have a significant impact on performance. This is discussed only in the RoBERTa paper which shows that downstream F1 scores get consistently reduced on average by $0.35\%$. Alternative common approaches to overcome the large amount of padding in many datasets are **"un-padding"** as in Effective Transformer [5] and sorted batching (SORT) as in Faster Transformer [21], lingvo [28] fairseq [22], and RoBERTa. However, for running efficiently on arbitrary accelerators, these approaches require substantial hardware-specific low-level code optimizations only available on GPUs. Further details are in Sections C [1] and 4.4.

Beyond language models, packing has been also present in other areas of machine learning, however with little to no exploration in the literature and mostly hidden in some libraries without any further discussion. For example, PyG (PyTorch Geometric) combines multiple small graphs in a batch to account for the large variation in size and to optimize the hardware usage when training a Graph Neural Network (GNN). Another example is the RNN implementation in PyTorch which introduces a "PackedSequence" object and states that "All RNN modules accept packed sequences as inputs" but does not address how sequences are packed efficiently and how the processing of packed sequences is implemented in an efficient manner while avoiding interaction between sequences. Even though we focus on BERT [6] and other transformers in this paper, the general principles can be transferred to many more machine learning algorithms with differently sized data samples.

In this paper, we formally frame the packing problem in transformer based models, and provide some solutions, showing that sequences can be packed efficiently, separator tokens are not required, and cross-contamination can be avoided with little overhead.

In summary, the contributions of the paper are as follows. In Section 2, we produce histograms of a variety of datasets showing the high percentage of padding tokens. In Section 3.1, we present two new deterministic and efficient packing algorithms based on established solvers which efficiently pack datasets with millions of sequences in a matter of seconds (or less). In Section 3.2 and Section 3.3, we describe 'cross-contamination' —the cause of the accuracy reduction which separator tokens do not mitigate— and show how the BERT model can be adjusted to show the same convergence behavior on packed and unpacked sequences. We empirically show that the proposed packing algorithms produce a nearly-optimal packing scheme for Wikipedia pre-training dataset (Section 4.1) and more in the Appendix. In Section 4.2, we demonstrate that the convergence of the BERT large model on the packed dataset is equivalent to that on the un-packed dataset with 2x throughput increase on the Wikipedia sequence length $512$ pre-training dataset. Further experiments underline the necessity and efficiency of our changes.

## 2   Sequence length distributions

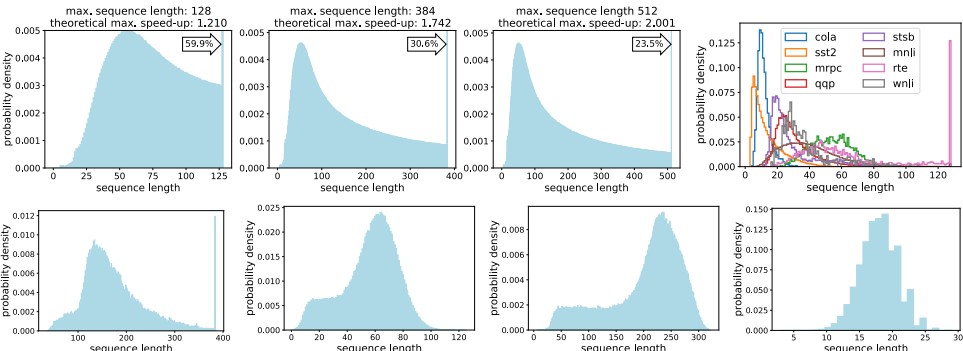

Figure 1: Sequence length distributions for different datasets. The three graphics at the top left show Wikipedia BERT pre-training dataset sequence length histograms (token count excluding padding) for different maximum sequence lengths based on the Wikipedia article dump from October 1st 2020. The theoretical speed-up relates to not using any padding tokens and not having any overhead from processing the different lengths. Top right: GLUE datasets. Bottom from left to right: SQuAD 1.1, LibriSpeech text labels, LibriSpeech audio token sequence, and QM9 molecules of a graph in a sequence.

BERT is pre-trained using masked-language modelling and next-sentence prediction on a large corpus of Wikipedia articles. Each sequence is composed of one <CLS> token followed by the first "segment" of sentences, followed by a <SEP> token, and then finally the second "segment" of sentences. Because these "segments" are created in sentence-level increments there is no token-level control of sequence length. Furthermore $10\%$ (default value, [7]) of sequences are intentionally cut short. This leads to significant levels of padding, especially for longer maximum sequence lengths (see Figure 1 and Section J[1]). At sequence length 128 (commonly used in phase 1 of pre-training) the theoretical speed-up is around 1.2, at sequence length 384 this increases to 1.7, and finally at sequence length 512 (commonly used for phase 2 of pre-training) it is 2.0. Despite the widespread use of the Wikipedia dataset for pre-training BERT such histograms have, to the best of our knowledge, not been published previously. This has perhaps lead to the underestimation of the speed-up opportunity available. To put things into perspective, the sequence length 512 dataset contains 8.33 billion tokens, of which 4.17 billion are padding tokens.

Note that the skewed sequence length distributions are neither limited to Wikipedia, as shown with GLUE [30, 31] from Section L[1] and SQuAD 1.1 [25] from Section K[1] ($2.2x$ speed up), to BERT training, as shown with LibiSpeech text distributions [23] from Section M[1], nor to text itself, given the LibriSpeech audio data distributions, and the QM9 molecular data [27, 26] ($1.6x$ speed-up, Section Q[1]). All distributions can be found in Figure 1. Since LibriSpeech audio data is skewed to longer sequences, only $1.3x$ speed-up could be achieved despite the theoretical maximum of $1.6x$. For all other cases, the algorithms presented in Section 3.1 lead to close to optimal packing.

# 3 Methods

Our approach consists of three distinct components. Firstly, we pack the $n$ data samples efficiently during pre-processing to make full use of the maximum sequence length, $s_m$ (Sections 3.1 and F). Secondly, we introduce a series of model changes in Section 3.2 that preserve the equivalence with the original BERT implementation. The changes include a self-attention mask to prevent the model from attending between different sequences in the same pack (Section 3.2.2), and an adjustment of the the positional embeddings (Section 3.2.1) to handle packs of sequences. Other components of the model, such as the feed-forward layer [29], operate on a per-token basis and do not require modification for pre-training. In Section 3.2.3, we also demonstrate how to compute a per-sequence loss and accuracy for NSP and downstream fine-tuning tasks. Thirdly, we provide suggestions for hyperparameter adjustment (Section 3.3) that lead to analogous convergence behavior between the packed and un-packed BERT implementations. Additional videos and animations are provided as supplemental material.

## 3.1 Packing algorithms

The widely studied and well established bin packing problem deals with the assignment of items into bins of a fixed capacity such that the number of utilized bins is minimized. It has been known for decades if not centuries. Since an exact solution is strongly NP-complete [14], numerous approximate solutions have been proposed [12, 15, 13, 36]. Since most existing approximations have a high complexity of at least $O(n \log n)$, we propose two new heuristic offline algorithms that are tailored to the NLP setting applied to the whole dataset. For a detailed introduction to packing see Section F.

### 3.1.1 Shortest-pack-first histogram-packing (SPFHP)

Shortest-pack-first histogram-packing (SPFHP) works on the bins in the sequence length histogram (with bin size 1) rather than the individual samples. The histogram is traversed in sorted order from longest to shortest sequences. Then, to pack the data during the traversal, we apply the worst-fit algorithm [12, 36] such that the histogram bin being processed goes to the **"pack"**[1] that has the most space remaining ("shortest-pack-first"). If the histogram bin does not fit completely, a new pack is created. We also limit the **packing depth**, in other words the maximum number of sequences that are allowed in a pack. Therefore, an existing pack is only extended if it is not already at maximum packing depth. The detailed code for the algorithm is provided in Listing 3. The time and space complexity of the algorithm are $O(n + s_m^2)$ and $O(s_m^2)$ (Section G.2[1]).

---

[1]We avoid the ambiguous terms "bin" and "sample/sequence"and use "pack" instead to refer to the multiple sequences concatenated during packing.

 ### 3.1.2 Non-negative least squares histogram-packing (NNLSHP)

The proposed NNLSHP algorithm is based on re-stating the packing problem as a (weighted) non-negative least squares problem (NNLS) [3] of the form $wAx = wb$ where $x \geq 0$. The vector $b$ is the histogram containing the counts of all the sequence lengths in the dataset. Next, we define the $A$ matrix (the "packing matrix") by first generating a list of all possible sequence length combinations ("strategies") that add up exactly to the maximum sequence length. We focus specifically on strategies that consist of at most 3 sequences per pack (independent of $b$) and encode each strategy as a column of the sparse matrix $A$. For example, a strategy consisting of the sequence length 128, 128, and 256 in represented a column vector that has the value 2 at the 128th row, the value 1 at the 256th row, and zero at all other rows. The variable $x$ describes the *non-negative* repetition count for each strategy. So a 24 in the $i$th row of $x$ means that the strategy represented by the $i$th column of $A$ should repeat 24 times. Moreover, in the un-weighted setting, $Ax = b$ states that we would like to "mix" the pre-defined strategies (columns of $A$) such that the number of samples matches the histogram $b$, and where each strategy is used $x \geq 0$ times. We use the residual weight $w$ to control the penalization of the $Ax - b$ residual on different sequence lengths (different rows of $b$). Heuristically, we set the weight of 0.09 for all sequences of length 8 or smaller because they are considered acceptable padding sequences while all other sequence lengths get weight 1. We discuss this heuristic choice of parameters in Section F.4.5 and F.5[1]. The overall efficiency of the packing is not greatly influenced by the weighing (less than 1% extra speed-up).

After solving $wAx = wb$ for $x \geq 0$ using an off-the-shelf solver, we obtain a floating point solution, which means that the repetition counts are not necessarily integers. Since we cannot use a non-natural number of strategies, we round the solution $\hat{x}$ to the nearest integer. The error introduced by this rounding is found to be negligible (a few hundred sequences in the worst case) compared to the size of the dataset (millions of sequences). The time complexity and space complexity of the algorithm are $O(n + s_m^5)$ and $O(s_m^3)$. Further details are provided in Section F.4.

## 3.2 packedBERT: model changes

This section describes how any vanilla BERT implementation should be modified for packed sequence processing, such that the behavior of the model is the same as when processing unpacked sequences. Preserving the mathematical equivalence is necessary to ensure existing BERT pre-training and fine-tuning practices remain valid, as well as being required by benchmarks such as MLPerf™ [17]. The presented approaches and principles apply to a variety of other models.

### 3.2.1 Adjust positional embeddings

The BERT model uses three types of embeddings: token, segment, and positional embeddings. The latter is canonically implemented as a bias add operation, rather than a full embedding look-up. This is possible because the positional indices increase linearly for every sequence. However, when using the packed data format the position index needs to be reset with each new packed sequence. For instance, when packing two sequences one of length 2 and one of length 3, the positional embedding indexes that need to be picked up are $[0, 1, 0, 1, 2]$. To achieve this, the bias add needs to be replaced by an embedding look-up to extract the correct positional embedding for each token in the pack. This also requires keeping an extra input which specifies the position of each token in its sequence. This required adjustment has only a minor impact on absolute accuracy/loss (see Section 4.2 and 4.2.1).

### 3.2.2 Adjust attention masking

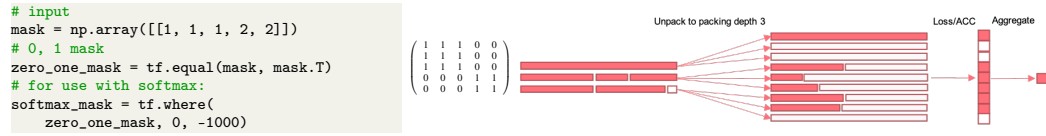

Figure 2: Attention mask code [left], respective zero-one mask [middle], and vectorized unpacking of the sequence loss[right]. White rectangles correspond to padding.

To maintain an implementation that is consistent with the un-packed version, tokens from different sequences within a pack should not be able to attend to each other. This is typically achieved in other implementations by unpacking the sequences using custom attention kernels and then doing the attention per-sequence [5]. Instead, we propose directly masking the attention matrix with a block-diagonal mask before the attention softmax. This is straightforward to implement in modern frameworks (see Figure 2). Naturally, there is a cost to both the mask construction and applying it to the attention matrix. However, it is required to keep the accuracy (see Table 1, Section 4.1, Section 4.2). See also the code of the deprecated tensor2tensor library and our own provided code.

### 3.2.3 Adjust per-sequence loss and accuracy

Canonical implementations of BERT compute the cross-entropy loss for the masked language model on a per-token basis. However other NLP tasks, such as SQuAD, compute the loss and accuracy on a per-sequence basis. This section discusses how to handle such tasks when training with packed sequences. Simply feeding packs of sequences to the same implementation of cross-entropy would result in a per-pack weighted loss. In other words, the overall loss on the micro-batch would sum-up the losses on the individual packs, rather than individual sequences. As a result, the model would converge to a different optimum than when running with the un-packed implementation. For instance, a pack of a single sequence would contribute to the loss with the same weight as a pack of three sequences.

To recover the per-sequence averaging behavior of the canonical un-packed BERT implementation, we effectively "unpack" the incoming logits and labels. Once the sequences have been unpacked, we can compute the loss on each sequence separately as usual and then add up the losses. However, rather than looping through the sequences index, we compute on all indexes in parallel (see Figure 2). This minimizes the latency overhead of un-packing the loss calculation. As an example, we show how per-sequence loss can be implemented for the pre-training task. We use the "masked lm weight" [7] input tensor to represent which sequence a given masked token belongs to (0, 1, 2 and so on). This is consistent with the canonical BERT implementation where this input takes a value of either 1 (belonging to the sequence) or 0 (belonging to padding). The full methodology is detailed in Listing 5 and can be applied to other classification or pre-training tasks.

## 3.3 Adjust hyperparameters

In terms of convergence behavior, the primary consequence of packing is an increase in the effective batch size (with respect to number of sequences and real tokens) with some added variation over different iterations. If we look on the sentence level, the number of sentences in one batch increases by the packing factor. Similarly, the number of tokens in one batch increases. Hence, hyperparameters that are sensitive to these numbers need to be adjusted.

A direct solution is to reduce the computational batch size by the packing factor (average number of sequences per pack) and keep all other hyperparameters the same. For example, if the packing factor is 2, cutting the gradient accumulation count by half is sufficient. The advantage of this strategy is that no fine-tuning of hyperparameters is required and performance curves are comparable. However, this approach might be not desirable as it might imply under-utilizing the memory/compute, especially if the micro batch size needs to be reduced.

Hence to preserve batch size and optimize hardware utilization, we additionally propose an approximate heuristic for updating the decay parameters of the LAMB optimizer [35] . For a packed dataset with a packing factor $p$, we update the decay parameters as: $\beta_1 := \beta_1^p$, $\beta_2 := \beta_2^p$. For $p = 2$, this corresponds to the exact parameters for calculating momentum and velocity, when updating with the same gradient twice (Section D). A common approach is to scale the learning rate with the batch size. However, our experiments in Section 4.2 show that this reduces convergence speed.

Since these adjustments are only heuristics the convergence of the model will be comparable but not identical. In particular, it is unlikely that simply adjusting the hyperparameters will fully undo the impact of the increased batch size. However, with these adjustments, researchers should be able to continue to use existing configurations.

## 4 Experiments

### 4.1 Bin packing algorithm comparison

We evaluate our algorithms using the following metrics: **number of packs**, **number of all tokens**, **number of padding tokens**, **solution time of the packing algorithm** (after histogram and strategy creation), **number of strategies used**, **packing efficiency** (the fraction of non-padding tokens in the packed dataset), the **speed-up** achieved compared to not packing (depth 1), and the average number of sequences per sample (**packing factor**). For SPFHP, we analyse different (maximum) packing depth, since packing is less efficient with smaller depth and we want to get a general understanding on how the packing depth influences the processing time. For NNLSHP, we focus on packing depth 3 because it packs the data sufficiently well. For the speed-up analysis, we focus on the intelligence processing unit (IPU) [11] (IPU-M2000, 16 accelerator chips), BERT phase 2 pretraining setup as in Section 4.2. A GPU dynamically loads the code into the accelerator; in contrast, the IPU works with a static pre-compiled engine that gets loaded onto the chip at the start of the run. While other approaches result in excessive padding or continuous changes of the code, our approach can work with the same code for the whole dataset. So in this setting the IPU architecture would especially benefit from our approach since it avoids code changes. Nevertheless, it can be applied to any implementation on GPU or TPU. For determining the speed-up, we take advantage of the precompiled kernel. Since time measurements are quite noisy, we can profile the kernel and how many cycles it takes for processing a batch. That way, we can determine the **overhead** (in cycles) from processing the additional attention masking and for unpacking the loss. Combining **overhead** and **packing factor**, we get the **speed-up** estimate. No experiment repetitions are required since the algorithms and measurements are deterministic.

Table 1: Key performance results of proposed packing algorithms (SPFHP and NNLSHP) on IPU.

| pack. depth | packing algorithm | EFF (%) | p | OH (%) | realized speed-up |
|---|---|---|---|---|---|
| 1 | NONE | 50.0 | 1.00 | 0.000 | 1.000 |
| 1 | SORT | 99.9 | 2.00 | $\gg$100 | $\ll$1.000 |
| $\approx$10 | GREEDY | $\approx$78 | $\approx$1.6 | $\approx$4.48 | $\approx$1.5 |
| 2 | SPFHP | 80.5 | 1.61 | 4.283 | 1.544 |
| 3 | SPFHP | 89.4 | 1.79 | 4.287 | 1.716 |
| 3 | NNLSHP | 99.7 | 2.00 | 4.287 | **1.913** |
| 4 | SPFHP | 93.9 | 1.88 | 4.294 | 1.803 |
| 8 | SPFHP | 98.9 | 1.98 | 4.481 | 1.895 |
| max | SPFHP | 99.6 | 1.99 | 4.477 | 1.905 |

**Packing depth** describes the maximum number of packed sequences. NONE is the baseline BERT implementation, whereas SORT corresponds to sorted batching, and GREEDY concatenates sequences as they arrive until they would exceed 512 tokens. Setting no limit resulted in a maximum packing depth of 16. **EFF**iciency is the percentage of real tokens in the packed dataset. The **p**acking factor describes the resulting potential speed-up compared to packing depth 1. With **overhead (OH)**, we denote the percentage decrease in throughput due to changes to the model to enable packing (such as the masking scheme introduced in Section 3.2.2). The **realized speed-up** is the combination of the speed-up due to packing (the **packing factor**) and the decrease in throughput due to the overhead on the IPU. It is used to measure the relative speed-up in throughput and the overhead from masking and loss adjustment. SORT can be only efficient on GPUs (see Section 4.4).

The main results for the performance metric evaluation are displayed in Table 1. The processing time for SPFHP on an Intel(R) Xeon(R) Gold 6138 CPU with 2.00GHz, 80 nodes, and $472G$ RAM was around $0.03s$ and independent from the packing depth. Classical First-Fit-Decreasing requires 87-120s, a lot of memory, and scales almost linear with the number of samples. We see that the overhead slightly increases with packing depth but that the benefits of packing outweigh the cost. The best speed-up is obtained with NNLSHP at depth 3 which required $28.4s$ on the CPU for processing and ran out of memory for larger depth. With a value of $1.913$, it is close to the theoretical upper bound of $2.001$. The results show that efficiency, packing factor, and speed-up can be viewed interchangeably. The amount of time needed to process a sample (a pack of sequences) is barely changed relative to the un-packed implementation. The packing factor, or the improvement in efficiency,

effectively provide an accurate estimate of the speed-up. GREEDY packing as used in T5 shows to be quite inefficient and sorted batching (SORT) is highly efficient in avoiding padding but the resulting different computational graphs cause a major overhead on the IPU that exceeds the benefits of avoiding the padding. Since we made our algorithm and code public available, results have been reproduced with a different framework on the Habana Gaudi accelerator [10] and confirmed that our approach is hardware and software independent giving it a huge advantage over existing approaches.

## 4.2  MLPerf™ phase 2 pretraining setup: learning curves and hyperparameter adjustment

For depth 1 (classic BERT) and NNLSHP with depth 3, we additionally evaluate on the MLPerf™ version 0.7 BERT pre-training benchmark [17]. Briefly, this involves training from a standard checkpoint to a masked-language model accuracy of $71.2\%$ using 3 million sequences with a maximum length of 512 tokens (refer to [19] for details). Following this standardized benchmark supports reproduction of results even on other systems and makes sure that the reproduction effort is moderate and setup rules are clearly documented. We compare the resulting speed-up as well as the respective learning curves by evaluating the data on a held-out validation dataset. The objective of this additional evaluation is to analyse if convergence behavior is changed by the packing strategy and if the theoretical speed-up can be achieved in practice.

With packing, we effectively increase the average batch size by the packing factor ($\approx 2$). However, with a different batch size, different hyperparameters are required (see Section 3.3) and there is no mapping that will generate exact matching of results but only heuristics. In a first comparison, we use the same hyperparameters when comparing packed and unpacked training except for cutting the accumulation count by half. This way, we make sure that the batch size is constant on **average** and we have the same amount of training steps. In the second comparison, we evaluate our heuristics and how they compensate the difference in batch size. This setup is more desirable because it is beneficial to use the hardware to its full potential and cutting the batch size by half usually reduces throughput. In the third comparison, we compare two optimized setups. In these two cases, packing takes half the amount of training steps.

The learning curves are displayed in Figure 3. In the first setup, we see the curves almost matching perfectly when normalizing by the numbers of samples processed. Differences can be explained by the variation of the number of sequences in the packing batch, and general noise in the training process. Especially after the initial phase, the curves show a near-identical match. The second setup shows bigger differences since changing the batch size and hyperparameters changes the training dynamics. We observe slower convergence early on in training due to the increased batch size. This is expected. The adjustment of the learning rate actually decreases performance probably because we correct for the increased number of sequences already in the modified loss. With the adjustment of the decay parameter of LAMB, we see matching performance at the later training stages. However, it is not feasible to completely recover the early convergence behavior of the smaller batch size by adjusting the hyperparameters. For instance doubling the batch size of unpacked BERT to 3000 and adjusting the LAMB decay parameters leads to more of a slow down in convergence than when running packed BERT with a batch size of 1500 and a packing factor of 2. n practice, our implementations exceeds the estimated 1.913 maximum speed-up. This estimate is based on the reduction in the computational work needed to process the dataset. However, packing the data also reduces the latency of the transferring the data to the device. Figure 3 shows that the realized total speed-up from packing exceeds $2x$.

### 4.2.1  Ablation study

So far, we have shown that with the introduced adjustments, we can match the accuracy of unpacked BERT. In the following, we analyze in how far the masking adjustment is required. In Figure 4, we can see that without our adjustments, training loss and accuracy worsen drastically and a longer training time does not lead to a recovery. When not adjusting the positional embedding, the loss and accuracy almost match. However, the accuracy stalls at $71.8\%$ and does not reach the target accuracy of $72.1\%$. So overall, both adjustments are crucial to avoid a reduction in performance.

When running packed BERT without the NSP loss but keeping everything else the same in a full training setup, we observed that downstream performance on SQuAD reduced the F1 measure by $1.31\%$ and EM by $1.15\%$. Hence, we do not consider removing NSP as done in approaches like RoBERTa and T5 as discussed in Section 1.

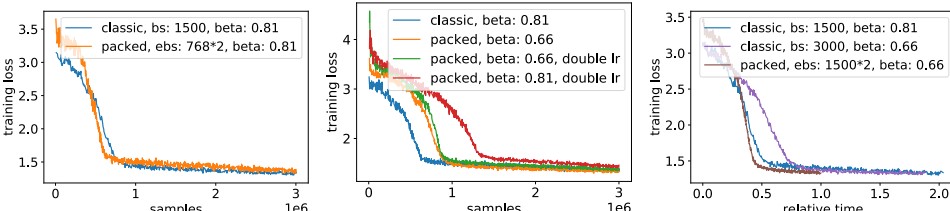

Figure 3: Comparison of learning curves for packed and unpacked processing, where all experiments converged to the target accuracy within the same number of training samples(3 million). [left] same **e**ffective **b**atch **s**ize (**ebs** is batch size times packing factor), [middle] different heuristic adjustments of the hyperparameters (batch size $1500$ for all runs, such that **ebs** for packed runs is $1500 * 2$), and [right] realized speed-up from packing (in excess of desired 2x). Further learning curves are provided in Section O.

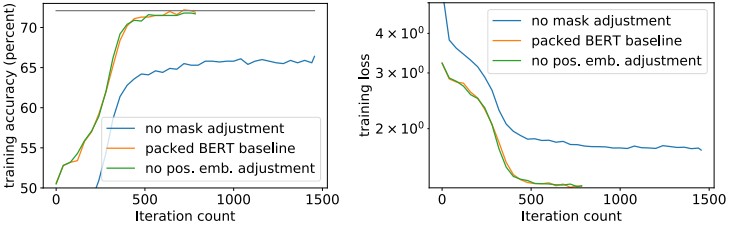

Figure 4: Comparison of learning curves with and without mask or positional embedding adjustment in our packed BERT approach. The grey accuracy baseline to reach is $72.1\%$.

### 4.3 Full pretraining and SQuAD finetuning

Packing slightly violates the i.i.d. assumption of data. Thus, we have to check that downstream performance is not impacted by packing. This is especially relevant in a full training setup without a starting checkpoint. To this aim, we show that the packed and unpacked SQuAD 1.1 scores are comparable after a full-pretraining of BERT base and large plus fine-tuning. During pre-training, in order to avoid giving an advantage to packing by further hyperparameter tuning, we reduce the gradient accumulation count for the packed BERT training for phase 1 and phase 2 to match, on average, the total number of sequences that get processed before each weight update. With this approach, we can use the same hyperparameters and number of training steps but process each batch faster by avoiding the processing of padding. This gives a slight disadvantage to the packed run in terms of machine utilization, as explained in Section 3.3 and is different to the speedup analysis in Section 4.2. For Phase 2, we use sequence length $384$ since longer range attention is not relevant for SQuAD 1.1. The respective speed-ups from packing for BERT base and large are shown in Table 2: the realized speed-up, measured as the quotient of the throughputs between the packed and unpacked runs, is slightly lower to the theoretical throughput (i.e. the packing factor) due to the packing overhead. Further learning curves with the loss function and accuracy are provided in Section P. For the fine-tuning training on SQuAD 1.1, we do not use packing. The scores, computed as the median of 10 different seeds, are displayed in Table 3. They are comparable to the reference ones in [6]: for BERT base (resp. large) the F1 score is reduced by $0.2\%$ (resp. $0.3\%$) and the EM score increases by $0.3\%$ (resp. $0.02\%$).

Table 2: Measured speed-ups in BERT pretraining with packing.

| Model size | Sequence length | Packing factor | Realized speed-up |
|---|---|---|---|
| base | 128 | 1.17 | 1.15 |
| | 384 | 1.70 | 1.68 |
| large | 128 | 1.17 | 1.15 |
| | 384 | 1.70 | 1.69 |

Table 3: SQuAD 1.1 scores after BERT pretraining with packing.

| Model size | Configuration | F1 | Exact match |
|---|---|---|---|
| base | [6] | 88.5 | 80.8 |
| | Packed | 88.32 | 81.03 |
| large | [6] | 90.9 | 84.1 |
| | Packed | 90.65 | 84.12 |

### 4.4 Scaling analysis: Impact of accelerators count

A further advantage of packing over competing un-padding approaches is the inherent load balancing provided by packing. So called un-padding approaches rely on dynamically launching custom kernels that ignore padding. A stated advantage of such implementations is the ability to avoid computing the complete (512 x 512) attention matrix. This provides additional computational savings compared to packing, where the attention matrix is computed in its entirety and then masked. Because of these additional savings, un-padding can exceed the theoretical upper bound for speed-up from packing (2.013 on Wikipedia). As a result of the dynamic nature of the approach, the processing time with un-padding is different for each sequence in the batch, and the amount of time required to process a batch of sequences will be determined by the processing time of the longest sequence in the batch (with the sequences being processed in parallel). Furthermore, in the multiple accelerator setting the processing time on each device will vary depending on the sequences in the batch that it receives. Devices which finish early have to wait for the slowest device to finish before exchanging gradients. This load-imbalance between the devices (and inside the batch) leads to a considerable decrease in the speed-up from un-padding as the number of accelerators is increased (see Figure 5 and Section E [1]). In contrast, packing (our approach) is inherently load-balanced. The processing time on each accelerator is independent of the content inside the batch received by the device. Any number of accelerators can therefore operate in unison without having to wait for the slowest batch to process (all per-device batches are equally fast).

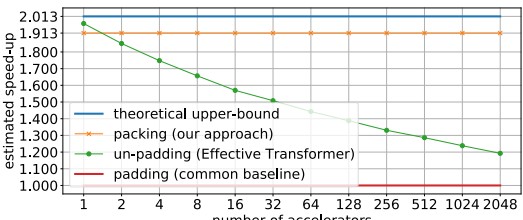

Figure 5: Comparison of the theoretical speed-up as the number of accelerators is increased.

## 5 Conclusion

Whereas packing is a well known concept, this paper sheds a new light onto it in multiple aspects. First, we visualize the sequence length distributions of multiple datasets not just from language domains but also audio and molecular domains to emphasize that packing is beneficial for a lot of datasets and that in many cases, more than 2x acceleration can be achieved by removing $50\%$ or more padding. Second, we provide two new highly efficient packing approaches based on established solvers that leave almost no padding and that can tackle arbitrarily large datasets in a matter of seconds, in contrast to existing approaches that are slow and suboptimal. Third, we demonstrate that without adjusting the sequence processing algorithm (e.g., BERT) to the packed sequences, predictive performance is reduced. Thus, we propose several model adjustments that are all necessary to keep predictive performance. Last but not least, we prove that, thanks to such adjustments, predictive performance is preserved as if no packing was used — but speed significantly increases, especially since the adjustments come with an overhead of less than $5\%$. We prove in our experiments that downstream performance is not impacted by packing and that the anticipated 2x acceleration can be achieved.

In the future, an interesting direction is the packing of images of different sizes to help accelerate computer-vision applications. This is especially relevant given the recent advances in the use of transformer-based approaches in the computer vision domain, for example the visual transformer [33]. Note that many images come in different shapes and resolutions and packing them can be a new approach to tackle this diversity instead of casting them all to the same resolution and shape. Masking out the self-attention within transformers is easier to implement than avoiding cross-contamination of convolutions applied to packed images. Future work should explore improving the performance of other models (RoBERTa, GPT-3, T5) by avoiding contamination between non-contiguous segments from different documents. Even BERT itself might benefit from avoiding contamination between the two concatenated segments.

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
