# Supplemental Material for "Efficient Sequence Packing without Cross-contamination: Accelerating Large Language Models without Impacting Performance"

**Table of Contents**

## A   Broader impact

We showed that when pre-training BERT on Wikipedia, the computational overhead taken to process padding tokens is roughly $50\%$. By eliminating this wasted computational time, the approach presented in this paper paves a way to halving the carbon footprint of training BERT-based models.

Furthermore, our approach circumvents the need for custom kernels, making the benefits of packing readily accessible to a broader audience of NLP practitioners. As such, we are hopeful the research will have a positive impact on the NLP community, and do not see any disadvantage of using this approach.

The benefit of our algorithm is based on two assumptions: A skewed length distribution in the training dataset and a hardware setup that trains efficiently on a fixed batch size. If *efficient training* is possible, with a variable batch size approaches like FasterTransformer and the fairseq sorted batch approach will result in the same or even larger benefits (due to smaller self-attention matrices). If the dataset is generated differently like in GPT models [4] and RoBERTa (FULL-SENTENCES) [16], all sequences will be at full length and sequences cannot be concatenated and there is indeed no benefit in packing sequences. However, strategies that reach full sequence length usually combine segments from different unrelated document sources which can result in reduced performance. Even in the normal BERT model, there might be this contamination between segments from different documents. Our paper introduced an approach to avoid the contamination between sequences. However, the same approach could also be applied to avoid contamination between segments and it remains future work to explore its benefits beyond BERT pretraining.

Future work would need to investigate the applicability of packing on text produced by different cultures and in different languages. We have already shown that the speed-up resulting from using our methods does not only occur when pre-training BERT on Wikipedia but also on other datasets such as SQuAD and GLUE. Furthermore, the sentence length distribution of the original English language text shows similar characteristics. Our research leads us to believe that compressible distributions arise naturally in language tasks and beyond, for instance in DNA sequence lengths [40], protein lengths [39], and speech (Section M). Many such sequence modelling workloads are based on variations of the BERT/transformer architecture and would therefore easily benefit from our acceleration.

Failures in NLP can have a big impact on society; many technologies, such as Alexa, Siri, and Google Home, rely on them. Whilst any errors arising from our approach can be avoided, one potential source of error comes from the implementation. Both the attention mask and the per-sequence loss need to be modified to support packing. These changes are significantly smaller than those required by custom kernels, however they may still be time consuming to implement and debug. To help mitigate the risk of any implementation errors, we share our reference implementations of the required changes in the appendix.

## B   Reproducibility Statement

All code for the packing algorithms is available in the appendix (Section T) and is directly linked to our GitHub page to simplify the download and usage. We even provide code for different variants and the histograms of sequence length for different datasets that got tokenized for BERT training of fine-tuning.

To generate the learning curves, our public submission to MLPerf™ could be used and we are preparing further code releases in other frameworks. To encourage the use of the adjustments of models for packed sequences, we additionally provide detailed explanations and code snippets in TensorFlow.

Detailed mathematical formulas (Section E and F), a theorem proof (Section D), and complexity calculations (Section G) are provided in this appendix to support our claims in the paper in full detail.

## C  Related work

The most obvious way to reduce the extent of padding in the dataset is to group samples by size before batching (SORT), i.e., process the shorter samples together and longer samples together. BERT is pre-trained in two phases, where the first phase uses sequence length 128 for 900K steps and the second phase uses sequence length 512 for 100K steps. However even by splitting the training in this way, the wasted compute due to padding is approximately 20% (see Figure 1). Other examples of this "sorted batching" approach can be found in Faster Transformer [21], lingvo [28] fairseq [22], and RoBERTa [16], which group samples of similar size together in one batch and fill up with padding only to the maximum length in this batch. This approach can be highly efficient in cases where the dataset length is multiple orders of magnitude larger than the batch size and the number of different sequence lengths. Despite its high computational efficiency, this approach has multiple drawbacks. We outline these below and propose an alternative which maintains the high efficiency, while also circumventing the downsides. Firstly, sorting the data can reduce the overall convergence speed when the batch size is large because it violates the i.i.d. assumption on the data distribution [2, 18]. Secondly, processing batches with shorter sequence lengths under-utilizes the compute compared to running the same batch size with a longer sequence length. For GPUs, a common heuristic to mitigate this effect is to adjust the batch size to keep the number of processed tokens near constant [22, 16]. In general however, the relationship between the sequence length and the optimum batch size is more complex and maximizing compute utilization can require the model to be sharded differently across multiple accelerators. Avoiding this, often manual process, is important for ease of use and the portability of methods across different hardware architectures. Thirdly, modern NLP applications are optimized and compiled for fixed tensor sizes using tools such as XLA [34, 9], which provides a $\approx 7x$ acceleration for BERT in MLPerf™ [17] compared to the non-XLA baseline [34]. Changing the sequence length or batch size requires re-optimization of the computational graph and recompilation of the program for the new tensor shapes. For complex models such as BERT, optimization and recompilation take a non-negligible amount of time. Even if one pre-compiled and cached all combinations of batch size and sequence length, the kernels would still need to be re-uploaded to the device every time the shapes change. Depending on how frequently the tensor shapes change, the overhead from switching kernels adds up. To avoid these issues, it is preferable (and common) to work with fixed tensor shapes for the entire duration of the training run.

More advanced approaches for reducing the padding overhead rely on custom computational kernels. Loosely these are referred to as **"un-padding"** approaches. In Effective Transformer [5], the input batch is provided as a padded matrix but padding values are dynamically removed and restored during different calculation stages. While un-padding implementations are highly sophisticated and are able to completely circumvent the processing of padding tokens, they introduce a significant overhead due to the multiple GPU kernel launches (i.e., one kernel per sequence rather than one kernel per batch). Additionally the time to process each batch will fluctuate depending on the sequence lengths in each batch, i.e., batches with only shorter sequences will typically be processed faster. When working with more than one accelerator, this variability in throughput results in all devices in the cluster waiting for the device with the most compute intensive batch to finish processing. As such, un-padding approaches are not appropriate for deployment on large clusters. The **"packing"** based approach introduced in this paper offers significant advantages over un-padding approaches. Firstly, packing is implemented directly at the framework level and requires no additional custom kernel implementations. Secondly, the processing time for each batch is independent of the content of the batch, allowing the packing based approach to maintain the same speed-up whether running on a single device or thousands.

While we demonstrate the effectiveness of packing specifically on the Wikipedia dataset, packing SQuAD [25] or GLUE datasets [31, 30] for BERT also leads to significant speed-ups (some in excess of 9x) (Sections K and L). The effectiveness of packing is a result of both the length distribution of the documents in the source datasets as well as the different text preprocessing steps for BERT [8]. The use of bi-directional self-attention in BERT implies that the input sequences should contain complete sentences. If a sentence is abruptly cut short, the hidden state on other (preceding) tokens in the sequence will be affected. Language models with causal attention (only attending to previous tokens in the input) do not have this issue to the same degree. For such models, if a sequence is cut short at an arbitrary token, the other tokens (which occur earlier in the sequence) will not be affected. This ability to cut sequences arbitrarily completely trivializes the packing problem for models based on causal attention. For instance, GPT-3 [4] is trained with a maximum sequence

length of $2048$ where a single sequence may contain multiple segments of sentences separated by a special end of segment token. The last segment in each sequence is simply cut to meet the sequence length requirement making the packing problem trivial and avoiding any padding. In the interest of computational efficiency GPT-3 does not mask the attention between different segments in a sequence. In contrast, the packing approach presented in this paper introduces a mask in the attention layer (see Section 3.2.2) to prevent cross-contamination between examples in a pack. Note, we mask the interaction between different sequences and not between different sentences or segments in the same sequence. This ensures that the characteristics of the original dataset and model are matched as closely as possible. RoBERTa and many other models in production like T5 [24] use a similar packing approach as GPT-3, combining full sentences/sequences with GREEDY packing (first come first concatenate) and also separation tokens or additional padding. The RoBERTa ablation study shows that mixing of sentences from different documents reduces accuracy, but it is used nonetheless for load balancing reasons which indicates that sorted batching is not sufficient.

There might be hidden code snippets as in the deprecated tensor2tensor library that seems to implement the same attention masking mechanism as we propose. However, these lack a sufficient documentation, testing, evaluation, ablation, and communication to the research community to be considered state of the art in NLP research. More general, to the best of our knowledge and the knowledge of many other engineers and researchers that we were in contact with, there is no other research work that focuses on packing in NLP.

# D  Theorem on LAMB hyperparameter correction heuristic

With packing, the effective batch size changes and hence hyperparameters of the LAMB optimizer [35] need to be adjusted. For a packed dataset with a packing factor $p$, we update the decay parameters as: $\overline{\beta_1} := \beta_1^p$, $\overline{\beta_2} := \beta_2^p$. For instance if $\beta_1 = 0.81$ for the un-packed dataset, then for a packed dataset with an average of $2$ sequences per sample one should use a value of $0.81^2 \approx 0.66$ instead. Assuming no or only minor changes in gradients and $p$ being a natural number, we can prove that this heuristic is the exact solution to make sure that momentum and velocity in LAMB are unaffected by packing. This can be proven by mathematical induction. Note that $p \geq 1$ by definition.

**Theorem D.1.** *For any $p \in \mathbb{N}$ and assuming that respective gradients on a batch of $b$ random samples are (approximately) the same, choosing*

$$\overline{\beta_1} := \beta_1^p \tag{1}$$

$$\overline{\beta_2} := \beta_2^p. \tag{2}$$

*as hyperparameters in the LAMB optimizer ensures that the momentum and velocity after $p$ separate update steps are the same as with one packed update step with $p \times b$ samples.*

*Proof.*

- *Base Case*:
  For $p = 1$ the left and right side of the equation are the same which matches exactly the unpacked case. Hence, the theorem holds for $p = 1$.

- *Inductive hypothesis*: Suppose the theorem holds for all values of $p$ up to some $k$, $k \geq 1$.

- *Inductive proposition*: The theorem holds for $p = k + 1$.

- *Proof of the inductive step*: Let $l$ be the loss function, $w_t$ the weight vector after $t$ updates, and $x_1^t, \ldots, x_b^t$ the respective underlying data to calculate the gradient $g_t$. For a single update step in LAMB with batch size $b$ samples, we compute the gradient

$$g_t = \frac{1}{b} \sum_{i=1}^{b} \frac{\partial l}{\partial w}(x_i^t, w^t). \tag{3}$$

Since $g_1 \approx g_2 \approx \ldots \approx g_{k+1}$, We have with the inductive hypothesis and the definitions in LAMB:

$$m_k = \beta_1^k m_0 + (1 - \beta_1^k) g_1 \tag{4}$$

$$v_k = \beta_2^k v_0 + (1 - \beta_2^k) g_1^2 \tag{5}$$

Now we can calculate (with $g_1 \approx g_{k+1}$)

$$m_{k+1} = \beta_1 m_k + (1 - \beta_1)g_{k+1} \tag{6}$$

$$\approx \beta_1 \left( \beta_1^k m_0 + (1 - \beta_1^k)g_1 \right) + (1 - \beta_1)g_1 \tag{7}$$

$$= \beta_1^{k+1} m_0 + (1 - \beta_1^{k+1})g_1 \tag{8}$$

The calculation for $v_k$ is the same. As reference for a packed update with $p = k + 1$ with $\overline{\beta_1}$ and $\overline{\beta_2}$, we would get

$$g = \frac{1}{pb} \sum_{j=1}^{p} \sum_{i=1}^{b} \frac{\partial l}{\partial w}(x_i^j, w^1) = \frac{1}{p} \sum_{j=1}^{p} \left( \frac{1}{b} \sum_{i=1}^{b} \frac{\partial l}{\partial w}(x_i^j, w^1) \right) \approx \frac{1}{p} \sum_{j=1}^{p} g_1 = g_1 \tag{9}$$

since we are calculating gradients over $b$ samples which are assumed to be approximately the same. Consequently, the updates for momentum and velocity would be

$$\overline{m_k} = \overline{\beta_1} m_0 + (1 - \overline{\beta_1})g_1 \tag{10}$$

$$\overline{v_k} = \overline{\beta_2} v_0 + (1 - \overline{\beta_2})g_1^2. \tag{11}$$

Hence, $\overline{\beta_1} = \beta_1^{k+1}$ and $\overline{\beta_2} = \beta_2^{k+1}$ is required to map to the formula with the consecutive updates (for the same amount of data).

- *Conclusion*: The theorem holds for any $p \in \mathbb{N}$.

$\square$

Since we proved that the formulas $\beta_1 := \beta_1^p$, $\beta_2 := \beta_2^p$. hold for all $p \in \mathbb{N}$, $p \geq 1$, it is safe to assume that it is an appropriate heuristic for all $p \in \mathbb{R}$, $p \geq 1$.

# E    Un-padding scaling estimate

To demonstrate the severity of the load-imbalance issue in Section 4.4 we consider the scaling of an un-padding approach with a per-device batch size of 32 running on eight devices [20]. From there, we readily extrapolate the performance to both larger and smaller cluster sizes by fitting a Gumbel distribution to the observed processing times as described in this section. On a single device with batch size 32 un-padding outperforms packing and exceeds the theoretical upper-bound for packing. As the number of devices increases to two or more, the proposed packing approach outperforms the dynamic un-padding approach. On a cluster with 32 accelerators the speed-up from un-padding drops to $50\%$ and with 2048 devices the speed-up is only $30\%$. In contrast, the speed-up due to packing is independent of the number of accelerators and stays at $1.913$. Switching to a smaller batch size would reduce the load-imbalance issue to some extent, but would also result in under-utilization of the available memory and compute.

Firstly, we retrieve the per-batch processing time for an un-padding implementation running pre-training on the Wikipedia dataset from [20]. These processing times were obtained using 8 GPUs each with a per-device batch size of 32. We also retrieve the throughput numbers for the same system running with padding from [44] and use that as the baseline to compare the un-padded throughput against.

The throughput on the 8 GPU system is effectively limited by the slowest of the eight batches being processed in parallel. The Gumbel distribution is particularly suited to modelling the maximum or minimum value of a fixed size collection of i.i.d. samples (in this case batches). We observe that on 8 GPUs the throughput (i.e. speed-up) distribution indeed closely resembles a Gumbel distribution with $\alpha_1 = 1.6$ and $\beta_8 = 0.13$ as shown in Figure 6.

We can extrapolate the performance on the 8 GPU system to larger clusters by recognizing that the processing time for each cluster is effectively determined by the slowest batch being processed. Specifically, we could randomly sample (without replacement) two processing times for the 8 GPU system, and record the max of the two as the processing time for a system of 16 GPUs. However, this simple approach is too sensitive to outliers in the data and would result in an under-estimate of the performance of un-padding on large systems. We mitigate the effect of outliers in the data

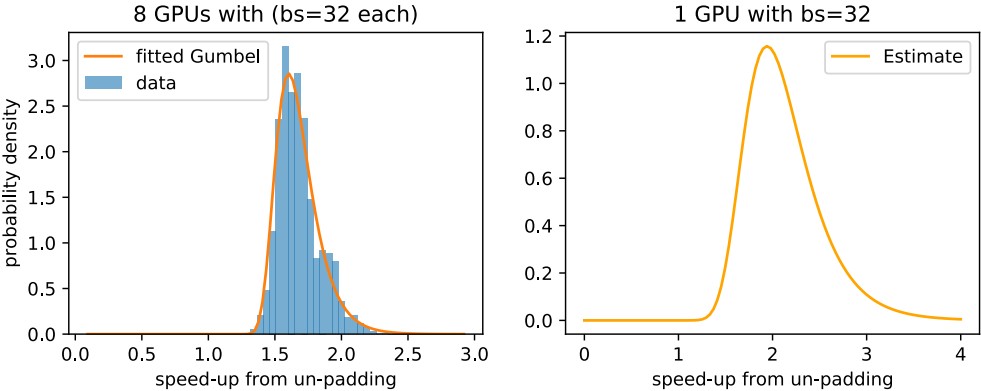

Figure 6: Left: Speed-up from un-padding on 8 GPUs closely resembles a Gumbel distribution. Right: statistical estimate of speed-up distribution on a 1 GPU system running un-padding

by avoiding directly sampling the processing times. Instead, we fit a Gumbel distribution to the processing times of a single batch of size 32 running on one GPU. To perform the fit, we observe that the cdf on one GPU ($P_1$) is related to the cdf on 8 GPUs ($P_8$) through [41](section 1.3):

$$(1 - P_8(s)) = (1 - P_1(s))^8 \tag{12}$$

In other words, if the speed-up on the cluster is larger than $s$, this implies that the speed-up on every GPUs in the cluster was at least $s$. Assuming $P_1$ is Gumbel and given the 8 GPU Gumbel parameters $\alpha_8$ and $\beta_8$, we need to fit two parameters, $\alpha_1$ and $\beta_1$. Consequently for the median ($s = \alpha_8 - \beta_8 \ln(\ln(2))$, $P_8(s) = 0.5$), we have:

$$0.5 = (1 - P_1(\alpha_8 - \beta_8 \ln(\ln(2))))^8 . \tag{13}$$

And since $P_8$ is Gumbel, we also have an equation for the mode ($s = \alpha_8$, $P_8(s) = e^{-1}$):

$$(1 - e^{-1}) = (1 - P_1(\alpha_8))^8 . \tag{14}$$

We solve these two non-linear equations simultaneously using the standard SciPy optimization package.

Listing 1: Infer Gumble distribution parameters.

```
import numpy as np
from scipy import stats, optimize
alpha_8 = 1.6038
beta_8 = 0.1288
def g(x):
    alpha_1, beta_1 = x
    dist = stats.gumbel_r(loc=alpha_1, scale=beta_1)
    # Equations for median and mode
    median = alpha_8 - beta_8*np.log(np.log(2))
    equation1 = 0.5 - dist.sf(median)**n_gpu
    mode = alpha_8
    equation2 = (1-np.exp(-1)) - dist.sf(mode)**n_gpu
    return (equation1**2 + equation2**2)

res = optimize.minimize(g, [alpha_8, beta_8], method="Nelder-Mead")
alpha_1, beta_1 = res.x
```

The resulting estimated speed-up Gumbel distribution for a single device has $\alpha = 1.94$, $\beta = 0.108$ and is shown in Figure 6 [right]. To simulate the performance of a cluster of size $n$ with a batch size of 32 per device, we take the minimum over $n$ samples from this distribution. Repeating this process to generate many samples allows us to estimate the expected speed-up for any given cluster size. Unfortunately, we cannot make any statistical inference about the processing times of individual sequences since the data is only provided at the granularity of 32 sequences per batch, and it is not clear how much of the computation is done in parallel and how much in serial.

# F Technical background on packing

## F.1 Canonical packing problem

The bin packing problem deals with the assignment of items into bins of a fixed capacity such that the number of utilized bins is minimized. In the canonical formulation of the packing problem a vector $s(i)$ of length $n$ is used to represent the items being packed, where $s(i)$ denotes the length of the i-th sequence/item. The allocation of items into bins is tracked through the assignment matrix $B$, where $B_{ij} \in \{0, 1\}$ states whether the i-th sequence should be placed into the j-th bin. In the worst case scenario, every item is assigned to its own bin, thus $B \in \mathbb{R}^{n \times n}$. Notably, $s$ grows linearly in the number of sequences/items being packed and $B$ grows with the square. To mask out unused bins $y_j \in \{0, 1\}$, denotes whether the j-th bin is being used. The optimization objective is to minimize the sum of $y_j$ while making sure to assign each $s_i$ to exactly one bin and not exceeding the maximum bin capacity $s_m$ for each bin. This problem formulation is well known as bin packing [14].

$$
\min_{y \in \{0,1\}^n, B \in \{0,1\}^{n \times n}} \quad \sum_{j=1}^{n} y_j \qquad \qquad \text{Minimize the number of bins.}
$$

$$
\text{s.t.} \quad \sum_{j=1}^{n} b_{ij} = 1 \quad \forall i \qquad \text{Assign each length/sequence to only one bin.}
$$

$$
\sum_{i=1}^{n} s(i) b_{ij} \leq s_m y_j \quad \forall j \qquad \text{Cumulative length cannot exceed capacity.}
$$

$$
\tag{15}
$$

Bin packing is a strongly NP-complete [14] problem. Producing an exact and optimal solution is possible with a variety of existing algorithms, for example with the branch-and-cut-and-price algorithm [37]. However, given that we want to apply it for very large $n$ (16M for the Wikipedia dataset) an approximate approach is required.

## F.2 Approximate bin packing problem

Approximate packing approaches are divided into online and offline algorithms [12]. Online algorithms process incoming sequences one-by-one in a streaming fashion, whereas offline algorithms have a holistic view of all samples to be packed but typically still operate on a per sample basis. This results in best case time and memory complexities of at least $O(n \log(n))$ and solutions that can sometimes be far from optimal, especially for the online algorithms which do not have access to a holistic view of the datasets. The simplest online approach (next-fit) would be to keep a single open bin at any given time. An incoming sequence is added to this open bin if it fits, otherwise the bin is closed (can never be appended to again) and a new one is opened to accommodate the new sequence [12]. In the case of the Wikipedia pre-training dataset almost 25% of the sequences are of length 512, which makes this approach very inefficient since bins would frequently be closed because the incoming sequence did not fit. More specifically, this approach is not able to efficiently combine one long sequence with one shorter sequence, when the number of long sequences is large. The algorithms that come closest to the approaches proposed in this paper are the online harmonic-k algorithm [15], which creates harmonic sized bins for the assignment decision, and the offline Modified First Fit Decreasing method [13, 36], which sorts the data, groups it into 4 size categories and defines a strategy adjusted to these sizes.

In our approaches, we make three major simplifications. We make the problem of bin packing less dependent on $n$ by operating on the histogram of sequence lengths with bin size 1. Hence, we replace $s(i)$ by its histogram $b$ and the bin assignment $y, B$ by a mixture of strategies $x$, where the set of all available packing strategies is modeled as the matrix $A$ (see also Section F.4.2).

Then, we do not solve the full packing problem but focus on a fixed packing depth (in other words the well known 3-partition problem). Last but not least, we solve the limited depth packing problem only approximately either with a non-negativity-constrained linear least squares [3] (NNLS) followed by rounding to nearest integer solution or by applying Worst-Fit [13, 36] to the histogram, sorted from largest to smallest (in contrast to using an unsorted dataset). An exact solution would not be appropriate, since the 3-partition problem is strongly NP-complete [38] as well.

## F.3 Definitions

In this section, we standardize the terms used throughout our methods. Firstly, the terms *pack* and *bin* may be used interchangeably. Secondly, the presented packing schemes impose a limit on how many sequences can be packed into any given bin. This limit is referred to as the maximum *packing depth*. For simplicity, we require the different sequence lengths in a pack to always add up exactly to the bin capacity $s_m$ (we can always generate a padding sequence of just the right length to fill-up the bin). A *packing strategy* is a sorted list of sequence lengths, for example $[5, 7, 500]$, such that the total sequence length is no more than $s_m$ and the number of sequences in the pack does not exceed the maximum *packing depth*. The output of a packing scheme is typically as set of *packing strategies* and the corresponding *repeat count* for each strategy stating how many times each strategy should be repeated in order to cover the entire dataset. The strategy *repeat count* is also referred to as the *mixture* of strategies. The objective of the packing algorithm is to jointly design a set of packing strategies and their repeat counts, such that the amount of *padding* is (approximately) minimized. The presence of *padding* in the packs can either be implicit or explicit. For instance for $s_m = 512$ the strategy [2, 508] has an implicit padding of 2 (needed to fill the pack up to the $s_m$). Alternatively, the strategy repeat count may over-subscribe a particular sequence length leading to explicit packing. For instance constructing a pack of [4, 508] may require a new *padding* sequence of length 4 be constructed, if there are not enough sequences of that length in the dataset. The packing algorithms, we present, use both representations.

## F.4 Non-negative least squares histogram-packing

The first algorithm proposed in this paper is suitable for settings where it is desirable to achieve a high packing efficiency with a limited packing depth. The algorithm is deterministic and has three major components described in Sections F.4.1, F.4.2 and F.4.3.

### F.4.1 Enumerating packing strategies of fixed packing depth

Listing all unique ways of packing up to a maximum *packing depth* can be achieved through dynamic programming. We only consider packing at most up to 3 sequences per pack. This is the smallest packing depth that can eliminate the need for most padding on the Wikipedia dataset. Increasing the depth to 4, increases the size of the packing problem drastically and yields no throughput benefit.[2] With only two sequences, packing would be not as efficient since the distribution on sequence length is not symmetric. We use dynamic programming to enumerate all feasible ways/strategies that up to $M$ sequences of length $1 - 512$ can be packed into a bin of length $512$. For example, a packing strategy may be $[512]$ or $[6, 506]$ or $[95, 184, 233]$. To avoid listing the same strategy multiple times, we enforce the sequence lengths within a pack to occur in sorted order, for example, $[95, 184, 233]$ is equivalent to $[184, 95, 233]$ and should only be listed once. This reduces the search space as well as the space of potential solutions by a factor of 6 approximately and thus significantly accelerates the optimization process. If you had the same strategy repeated 6 times instead of having just one instance of that strategy with weight $X$, you will have six instances with weight $x/6$ (for example, or any other distribution). This would conflict with integer rounding of the solutions and with convergence of optimization algorithms.

### F.4.2 Constructing the packing matrix

The number of rows in the packing matrix is equal to the number of different sequence length categories. For instance, if we are using a granularity of 1 token to distinguish between different sequence lengths, then there are "maximum sequence length" rows. Each column of the matrix corresponds to a valid packing strategy (given the depth of packing). An example packing matrix for fitting up to 3 sequences into sequence length 8 is given in Table 4. Each column of the matrix represents a packing strategy. For instance, the first column represents the strategy [1, 1, 6] of packing two length-1 sequences and one length-6 sequence together to form a pack of length 8. The number of strategies (and columns in the matrix) is discussed in Section G. For a packing depth of 3 and maximum sequence length, we obtain around $\frac{s_m^2 + 6s_m + 12}{12}$ strategies. For depth 4, around $\frac{s_m(s_m+4)(2s_m+1)}{288}$ more get added.

---

[2] For data distributions that are more skewed than Wikipedia this might look different.

Table 4: Example packing matrix for sequence length 8. Columns represent different kinds of packs. Rows represent the number of sequences in these packs with a certain length. The last column represents a pack with only a single sequence of length six.

| 2 | 1 | 1 | 1 | 0 | 0 | 0 | 0 | 0 | 0 |
|---|---|---|---|---|---|---|---|---|---|
| 0 | 1 | 0 | 0 | 2 | 1 | 1 | 0 | 0 | 0 |
| 0 | 0 | 1 | 0 | 0 | 2 | 0 | 1 | 0 | 0 |
| 0 | 0 | 1 | 0 | 1 | 0 | 0 | 0 | 2 | 0 |
| 0 | 1 | 0 | 0 | 0 | 0 | 0 | 1 | 0 | 0 |
| 1 | 0 | 0 | 0 | 0 | 0 | 1 | 0 | 0 | 0 |
| 0 | 0 | 0 | 1 | 0 | 0 | 0 | 0 | 0 | 0 |
| 0 | 0 | 0 | 0 | 0 | 0 | 0 | 0 | 0 | 1 |

### F.4.3 Solution of the NNLS approximate packing problem

A solution of the packing problem is the mixture of packing strategies $x$ that minimizes the amount of padding in the packed dataset. We solve directly for the mixture (positive real numbers) and recover the padding as the negative portion of the residual (see Section F.4.4).

$$\min_{x \in \mathbb{R}^m} \quad \|A \cdot x - b\|^2$$
$$\text{s.t. } x \geq 0 \tag{16}$$

The solution vector $x$ will represent the mixture of the columns of $A$, in other words the mixture of valid packing strategies such that $A \cdot x$ is as close as possible (in the least squares sense) to the histogram of sequence lengths $b$. We obtain a solution with a non-negative least squares implementation [42, 46] Interestingly in the case of sequence length 512 only 634 out of the 22102 available packing strategies of depth up to 3 are used (3%).

### F.4.4 Padding as the residuals of the packing problem

We compute the residuals of the least squares solution (after rounding the mixture to integer) as:

$$r = b - A \cdot round(x) \tag{17}$$

The negative portion of the residuals represents sequences that we are "short". That is, there is a deficit of those sequences and we are over-subscribing to them. The positive portion of the residuals represents sequences which have failed to be packed. Typically, there is a deficit of short sequences and a surplus of long sequences as demonstrated by the following plot.

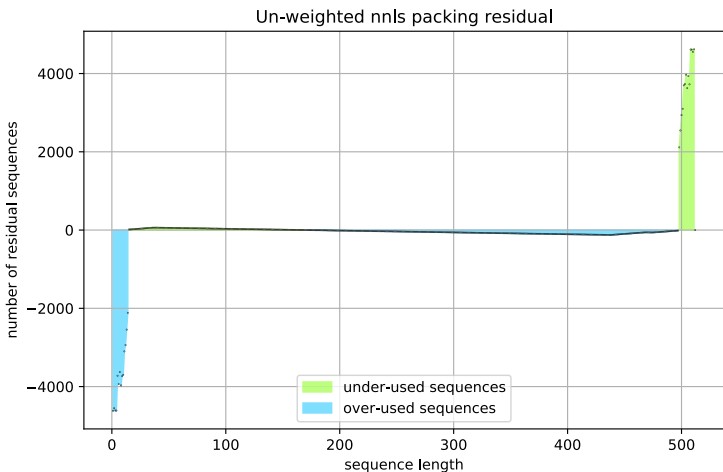

Figure 7: Visualization of the residual of the NNLS packing problem

In total, there are $n = 16\,279\,552$ sequences in the Wikipedia pre-training dataset. After the non-negative least squares packing (and rounding to integer solution) there are $56\,799$ un-packed sequences left un-packed (about $0.352\%$). The residual on sequence lengths 1 to 8 are

$[-4620, -4553, -4612, -4614, -3723, -3936, -3628, -3970]$. These negative residuals imply that we need to add this many sequences of their corresponding sequence length to realize the mixture of packing strategies. In total the first iteration introduces $7.9410^6$ tokens of padding. In contrast large sequence lengths have a positive residual (a surplus of unused sequences). For sequence lengths 504 to 512 the values are $[3628, 3936, 3724, 4613, 4612, 4553, 4619, 0]$. Note that sequence length 512 has a residual of 0 since they do not need packing. Intermediate sequence lengths typically have non-zero (but much smaller) residuals.

The detailed code for the algorithm is provided in Listing 2.

### F.4.5 Residual weighting

A natural extension of the non-negative least squares problem introduced in Section F.4.3 is to weight the residuals on different sequence length differently.

$$\min_{x \in \mathbb{R}^m} \quad \|(wA) \cdot x - (wb)\|^2$$
$$\text{s.t.} \quad x \geq 0 \tag{18}$$

We should not significantly penalize a deficit in short sequence lengths (smaller than 8 tokens) as adding up to 8 tokens of padding is not much overhead. Similarly, a surplus in long sequences is not worrisome because the amount of padding needed to achieve a sequence length of 512 is small. Reducing the weight of the residual on the first 8 tokens to 0.09 leads to the following residual plot shown on the right in Figure 8. In this case the residual is almost entirely shifted to the shorter sequences and the positive residual on the longer sequences has virtual disappeared.

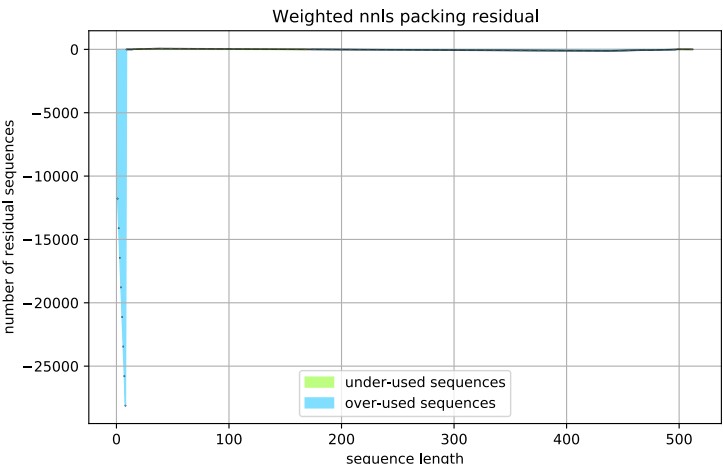

Figure 8: Visualization of the weighted residual of the NNLS packing problem

## F.5 Discussion of residual weight choice

This section discusses the choice and effect of the weighting parameters in the NNLSP packing algorithm. To simplify the problem of selecting reasonable defaults for the residual weights, we use just two parameters to completely describe the weights: an "offset" parameter and a "weight" parameter. Originally, all sequence length residuals are given the same weight of 1. This results in a packing with leftover long sequences, because there are not enough short sequences to pack them with. To reduce the residual on long sequences, we could either increase the residual weight on long sequences or reduce the weight on short sequences. We chose to reduce the weight on short sequences. Specifically, sequence lengths up to the "offset" length have a reduced "weight". The other residual weights stay at 1.

To start, we chose an offset of 8 tokens, which is the smallest power of 2 for which there are examples in the Wikipedia dataset. We decrease the weight on sequences shorter than the "offset" from 1 to 0.9 to 0.09 to see which order of magnitude is the most appropriate. On visual inspection (looking at the residual plots as in Figure 8), we found that 0.9 still left too many long sequences unpacked. So, we reduced the weight a further order of magnitude to 0.09. This seemed sufficient to encourage nearly all long sequences to pack. While visual inspection helps in understanding how many long/short sequences are leftover, we are also interested in the impact the weights have on the overall efficiency of the packing.

Without any weighting, we get 99.746359% efficiency, whereas the weighted approach results in 99.746274% efficiency. Hence, we conclude that the impact of the weights on the packing efficiency is very limited. Additionally, using an "offset" length of 4, resulted in similar numbers, for the full range of weights from 0 to 1. Using a weight of 0 for an "offset" length of 8 resulted in insignificantly higher efficiency of 99.7519%, whereas using an "offset" length of 16 reduces performance to 99.38964%. A weight of 0 implies that the residual on those lengths can be safely ignored, i.e., the packing algorithm can thus add as many short sequences as it chooses without any penalty. It is very interesting that this does not significantly impact the packing efficiency, and can even have a slightly positive impact. However, increasing the "offset" length further significantly decreases the performance with weight 0. Keeping the weight at 0.09 and increasing the length reduces performance slightly, for example with 99.53% at length 256 and 99.728% at length 16.

For our Squad analysis, weighting improved the efficiency slightly from 96.94% to 97.38%. Fine tuning further with direction grid search, delivered a local optimum of 98.767% efficiency with length 64 and weight 0.002.

Overall the influence of different residual weights on the packing efficiency (and the acceleration factor) is less than 1%. This might differ from application to application, but it shows that we are able to use the residual weights to achieve secondary targets (like not having leftover long sequences) without significantly compromising the packing efficiency.

## G  Complexity analysis of the proposed packing approaches

Since approximate packing algorithms have a complexity of at least $O(n \log(n))$ and we would like to be able to tackle datasets with 2K million samples, we will discuss the complexity of our packing algorithms in this section. The complexity depends on the maximum sequence length $s_m$, the number of samples $n$, and the packing depth $d$.

To create the histogram, we have to iterate over the data once ($O(n)$). Our histograms will be binned by size 1, meaning one bin for each sequence length. Hence, a dictionary can be generated ($O(s_m)$) and used for the sorting ($O(1)$). The respective histogram vector has dimension $s_m$.

### G.1  Complexity Analysis of non-negative least-squares histogram-packing

For a packing depth of one, there is only the strategy $[s_m]$. For a packing depth of two, we add the strategies $[1, s_m - 1], ..., [s_m - \lfloor \frac{s_m}{2} \rfloor]]$ which results in an additional $\lfloor \frac{s_m}{2} \rfloor$ potential strategies. Following the dynamic programming approach, the number of possible additional strategies of depth three can be calculated with

$$
\begin{aligned}
\text{\# potential strategies} &= \sum_{j=1}^{\lfloor \frac{s_m}{3} \rfloor} \sum_{i=j}^{\lfloor \frac{s_m-j}{2} \rfloor} 1 = \sum_{j=1}^{\lfloor \frac{s_m}{3} \rfloor} \left\lfloor \frac{s_m - j}{2} \right\rfloor - (j-1) \\
&\approx \sum_{j=1}^{\lfloor \frac{s_m}{3} \rfloor} \frac{s_m}{2} - \frac{3}{2} j \approx \frac{s_m}{2} \frac{s_m}{3} - \frac{3}{2} \frac{s_m/3(s_m/3+1)}{2} \\
&\approx \left\lceil \frac{s_m^2}{12} \right\rceil
\end{aligned} \tag{19}
$$

Note that for $s_m = 512$ the approximation is exact. This means that our strategy matrix $A$ has the dimensions $s_m \times \left( \left\lceil \frac{s_m^2}{12} \right\rceil + \lfloor \frac{s_m}{2} \rfloor + 1 \right)$. Overall, this leaves us with a space complexity of $s_m^3$ since $A$ is larger than $w$, $x$, and $b$. So it contains 11'316'224 numbers which is still much smaller than $n$. Note that the original data matrix $B$ had $n^2$ entries, which all needed to be optimized together with the $n$ bin assignments $y$. We now have only $\left\lceil \frac{s_m^2}{12} \right\rceil + \lfloor \frac{s_m}{2} \rfloor$ free variables in the strategy vector $x$. Also note that $A$ can be precomputed when $s_m$ is known and is independent of the number of samples. Given a problem matrix with dimension $i \times j$, Luo et al. [43] indicate that the asymptotic complexity of most solution approaches is $O(ij^2)$, whereas they propose an $O(ij)$ solution. Since we use the standard SciPy implementation [42], our estimated total time complexity for NNLSHP is $O(n + s_m^5)$.

For $s_m = 2048$, the estimate would be $350'540$ potential strategies which is still far less than the number of samples. For packing depth 4, we calculate [48]:

$$
\begin{aligned}
&\sum_{k=1}^{\lfloor \frac{s_m}{4} \rfloor} \sum_{j=k}^{\lfloor \frac{s_m-k}{3} \rfloor} \sum_{i=j}^{\lfloor \frac{s_m-j-k}{2} \rfloor} 1 \\
&\approx \sum_{k=1}^{\lfloor \frac{s_m}{4} \rfloor} \sum_{j=k}^{\lfloor \frac{s_m-k}{3} \rfloor} \frac{s_m - k + 2 - 3j}{2} \\
&\approx \sum_{k=1}^{\lfloor \frac{s_m}{4} \rfloor} \frac{1}{12} (s + 4 - 4k)(s + 3 - 4k) \\
&\approx \frac{1}{288} s(2s^2 + 9s + 4) \\
&= \frac{1}{288} s(s + 4)(2s + 1)
\end{aligned} \tag{20}
$$

So with $s_m = 512$, there would be around $940K$ strategies. In our implementation, this number of strategies would be too high to create the problem matrix. One alternatives to simplify would be to

not use the exact length of sequences but to only consider even numbers for the sequence length and round up. That way arbitrary sequence length could also be handled and the limiting factor would be the complexity of the attention layer in BERT which does not scale well with the sequence length.

### G.2 Complexity Analysis of shortest-pack-first histogram-packing

The complexity calculation of SPFHP is straightforward. We go over the whole data once for the histogram sorting. Next, we iterate over each of the $s_m$ bins in the histogram. Lastly, we iterate over all strategies that were encountered so far. It can be proven that, at each iteration, the number of strategies can be maximally increased by one. In each step, we potentially add a sequence to existing strategies but a new strategy is opened up only in the final step, when we either create a new strategy or we split one of the existing strategies into two. Hence, the number of strategies is bounded by $s_m$ and the overall time complexity is bounded by $O(n + s_m^2)$. The space complexity is $O(s_m^2)$ since we need to store up to $s_m$ strategies with maximum $s_m$ counts for different sequence length.

## H   Performance Comparison to GREEDY Packing in T5

T5 [24] is normally trained on the C4 dataset. However, to give an idea of the difference in packing efficiency and acceleration compared to our newly introduced algorithm, we can analyse the performance of greedy aggregation of samples on our given Wikipedia dataset.

We take the histogram and cast it back to a list of different sequence lengths since this is all that matters for analysing packing behaviour. Next, we randomly shuffle the dataset and iterate with the greedy aggregation algorithm multiple times to account for randomness. We iterate sequence by sequence and combine them provided the maximum sequence length of $512$ is not yet reached. If it is exceeded, the packed sequence is considered finished and a new sequence is started.

The greedy packing algorithm itself takes a bit more than $10$ seconds, since we are operating on single sequences and not histogram counts. The efficiency of this approach is $78.24\%$ (standard deviation of $0.005$) compared to our $99.75\%$ for NNLSHP. The respective acceleration would be around $1.566x$ compared to our $2x$. With respective separator tokens, the performance decreases around $0.13\%$ for one separator token and $0.27\%$ when two separator tokens are required between two sequences. Following the brief documentation at tensor2tensor [link], two separator tokens would be expected in the T5 processing.

In addition to the packing preprocessing, our paper proposes, rather than using separator tokens, to instead modify the masking of the attention matrix during training. The RoBERTa paper shows that avoiding contamination of sequences from different documents can consistently improve downstream F1 scores by $0.35\%$.

## I   Impact of NSP loss

When running packed BERT base without the NSP loss but keeping everything else the same, we observed that downstream performance on SQuAD reduced the F1 measure by $1.31\%$ and EM by $1.15\%$.

For the packing in approaches like RoBERTa or T5, it is crucial that there is no NSP loss because that would circumvent putting arbitrary sequences together in contrast to our approach that can handle multiple sequences from different documents without cross-contamination. Liu et al. [16] argument that NSP can be omitted because "removing the NSP loss matches or slightly improves downstream task performance". In their experiments, they compare the normal BERT setup with NSP ("SEGMENT-PAIR") to the "DOC-SENTENCES" approach, where there is no NSP and data in one sequence comes only from one document. For the "SEGMENT-PAIR" approach, the paper does not address, how much padding tokens are still present. Assuming, it is around $40\%$, their correction in batch sizes for each step would result in a significant increase in training steps for the "DOC-SENTENCES" approach. It is well known that BERT performance increases with longer pretraining time. Our results indicate that NSP loss might be still relevant, depending on the dataset generation process. With our approach, we can get the acceleration benefits of T5 and RoBERTa while keeping the predictive performance by avoiding cross-contamination.

## J   Wikipedia with Longer Sequence Length

The histogram raw data for Wikipedia with different maximum sequence length is provided in Listing 6 and visualized in Figure 9. We can see that with increasing maximum sequence length, long sequences become more and more rare and the resulting benefits from packing drastically increase. Keeping in mind that the BERT dataset generation process decreases the size of a maximum of 50% of the sequences, we can infer that having a different dataset generator that truncates any short sequence, would result in significant loss of data ($> 25\%$ for length 512).

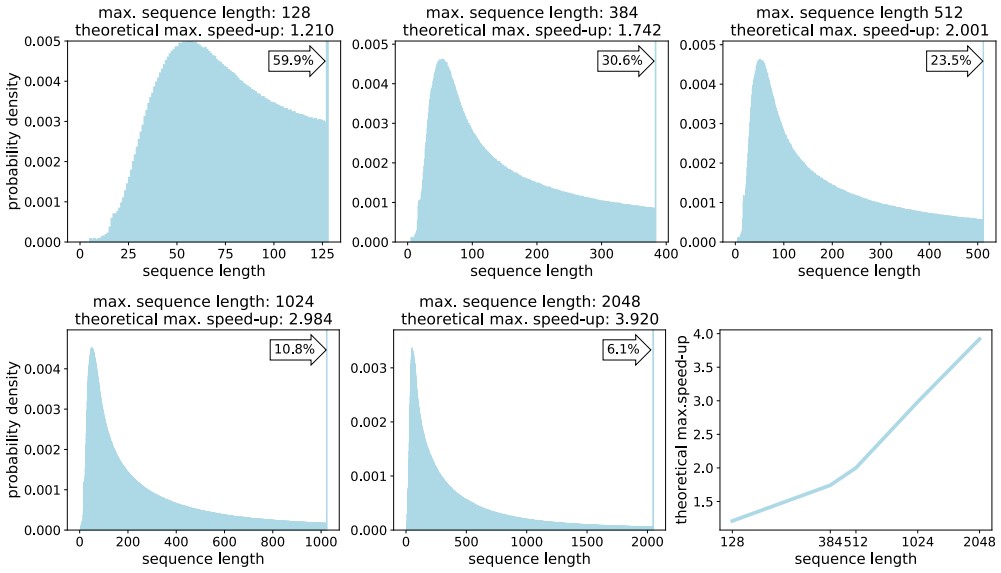

Figure 9: Sequence length distributions for different sequence lengths in Wikipedia BERT pre-training dataset and according theoretical speed-up.

Due to the length distribution, it is not anymore sufficient to concatenate only 3 sequences to obtain perfect packing for maximum sequence length 1024 or 2048. Instead, around 6 and 12 sequences are required. This cannot be solved by NNLSHP anymore due to search space complexity but requires an online heuristics like SPFHP or the slightly better LPFHP, introduced in Section R that is based on Best-Fit and splitting counts in the histogram in contrast to the rather simple First-Fit descending. Figure 10 shows the achieved speed-ups with LPFHP depending on the maximum number of allowed sequences.

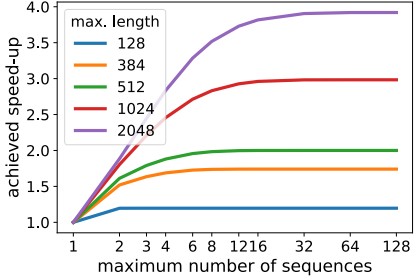

Figure 10: Speed-ups achieved by LPFHP for different maximum sequence length and maximum number of packed sequences.

## K   Packing SQuAD 1.1

We tokenized SQuAD [25] for BERT [6] with maximum sequence length 384 and visualized the histogram over the sequence length (Figure 11). The distribution looks similar to the Wikipedia dataset but is slightly less skewed. However, the maximum sequence length only had an occurrence of 1.2% compared to 23.5%. Hence, the theoretical un-padding speedup is 2.232. In Table 5, we can see that SPFHP does not concatenate more than 3 samples and obtains 97.54% efficiency in contrast to a maximally used depth of 16 with 99.60% efficiency on Wikipedia, because of the less skewed distribution. Note that we have less than $90'000$ samples. Hence, NNLSHP is less efficient because the rounding in the residuals has a much larger impact compared to more than 16 million sequences in the Wikipedia dataset.

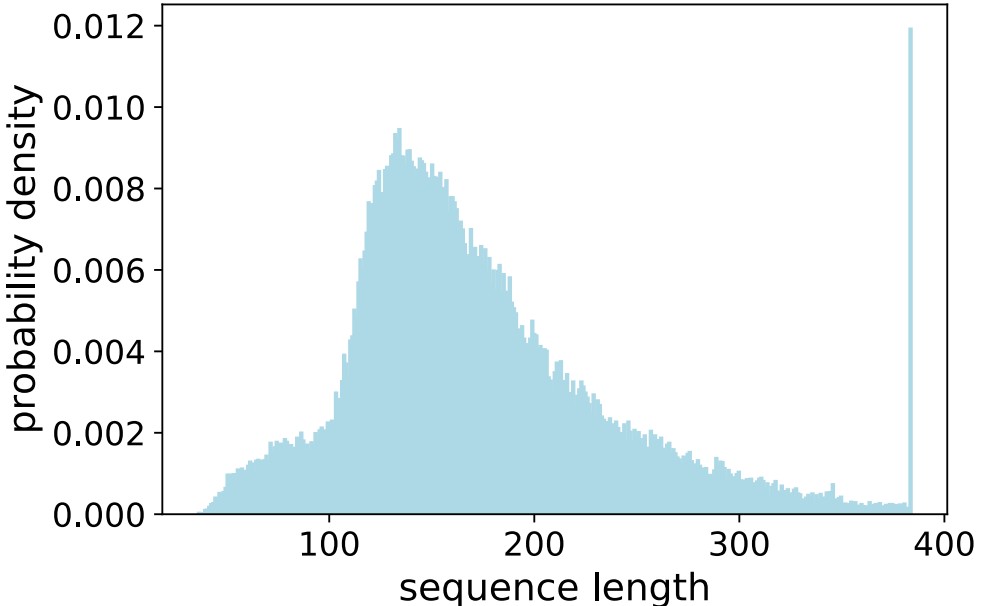

Figure 11: SQuAD 1.1 BERT pre-training dataset sequence length histogram for maximum sequence length of 384.

Table 5: Performance results of proposed packing algorithms for SQuAD 1.1 BERT pre-training.

| packing depth | packing algorithm | # strategies used | # packs | # tokens | # padding tokens | efficiency (%) | packing factor |
|---|---|---|---|---|---|---|---|
| 1 | none | 348 | 88641 | 34038144 | 18788665 | 44.801 | 1.000 |
| 2 | SPFHP | 348 | 45335 | 17408640 | 2159161 | 87.597 | 1.955 |
| 3 | NNLSHP | 398 | 40808 | 15670272 | 420793 | 97.310 | 2.172 |
| 3/max | SPFHP | 344 | 40711 | 15633024 | 383545 | 97.547 | 2.177 |

## L   Packing GLUE

To explore a variety of datasets and emphasize that skewed distributions are common, we explored all datasets in the GLUE benchmark [31, 30] that came with training data. We loaded the datasets using the HuggingFace dataset loading API [47]. For preprocessing, we followed the implementation in the HuggingFace transformers repository [32] [3] and extracted the respective data processing snippets to obtain tokenized data with a maximum sequence length of 128. The histogram of the sequence length for each of the included datasets is displayed in Figure 12 and the packing results are given in Table 6. Each dataset benefits from packing. The lower the mean, the higher the packing factors are that can be reached but with a higher packing depth.

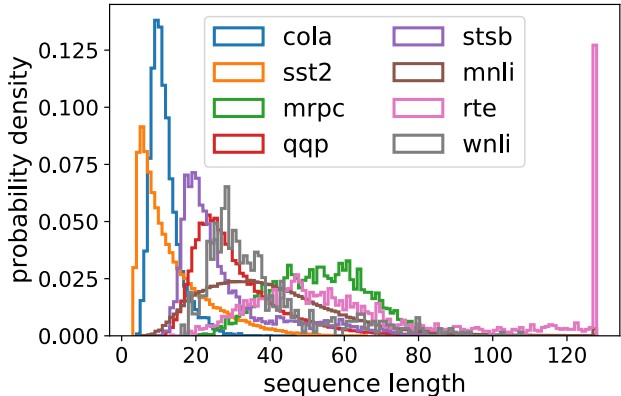

Figure 12: GLUE dataset sequence length histograms for maximum sequence length of 128.

Table 6: Performance results of proposed packing algorithms for the GLUE dataset. Only the baseline and the SPFHP packing results without limiting the packing depth are displayed.

| data name | packing depth | # strategies used | # packs | # tokens | # padding tokens | efficiency (%) | packing factor |
|---|---|---|---|---|---|---|---|
| cola | 1 | 34 | 8551 | 1094528 | 997669 | 8.849 | 1.000 |
| cola | 13/max | 29 | 913 | 116864 | 20005 | 82.882 | 9.366 |
| sst2 | 1 | 64 | 67349 | 8620672 | 7723633 | 10.406 | 1.000 |
| sst2 | 15/max | 64 | 7691 | 984448 | 87409 | 91.121 | 8.757 |
| mrpc | 1 | 77 | 3668 | 469504 | 274214 | 41.595 | 1.000 |
| mrpc | 4/max | 74 | 1606 | 205568 | 10278 | 95.000 | 2.284 |
| qqp | 1 | 123 | 363846 | 46572288 | 35448844 | 23.884 | 1.000 |
| qqp | 5/max | 123 | 97204 | 12442112 | 1318668 | 89.402 | 3.743 |
| stsb | 1 | 85 | 5749 | 735872 | 575993 | 21.726 | 1.000 |
| stsb | 6/max | 83 | 1367 | 174976 | 15097 | 91.372 | 4.206 |
| mnli | 1 | 124 | 392702 | 50265856 | 34636487 | 31.093 | 1.000 |
| mnli | 8/max | 124 | 123980 | 15869440 | 240071 | 98.487 | 3.167 |
| rte | 1 | 112 | 2490 | 318720 | 152980 | 52.002 | 1.000 |
| rte | 4/max | 108 | 1330 | 170240 | 4500 | 97.357 | 1.872 |
| wnli | 1 | 72 | 635 | 81280 | 57741 | 28.960 | 1.000 |
| wnli | 6/max | 63 | 192 | 24576 | 1037 | 95.780 | 3.307 |

---

[3] `https://github.com/huggingface/transformers/blob/master/examples/text-classification/run_glue.py`

## M   Packing Audio Data (LibriSpeech)

In this section, we show that packing can benefit other domains than NLP like ASR. We use the LibiSpeech dataset [23] and preprocess it as described at a reference implementation.[4] The resulting histograms for the subsampled audio sample lengths and respective text labels are provided in Figure 13

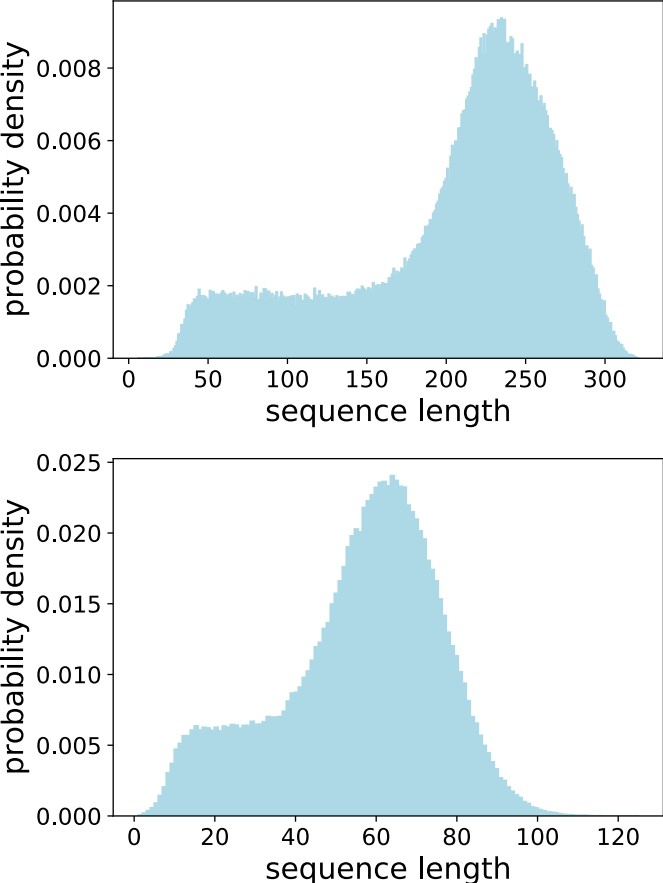

Figure 13: LibriSpeech sequence length histograms of preprocessed audio data [top] as well as target text data [bottom].

It can be seen that the audio sequence length is dominated by long sequences with $38\%$ of required padding to meet the max sequence length of $330$. Thus the theoretical optimal speed-up of $1.6x$ cannot be reached. However, $80\%$ efficiency are possible with any of the proposed packing algorithms to achieve $1.3x$ speed-up. This can be already achieved by combining up to 2 sequences. To achieve almost perfect packing efficiency, a sequence length around $457$ and concatenating up to $8$ sequences is required. Due to the quadratic increased computational load that usually comes with longer sequence length, increasing the sequence length is not practical.

If processing and packing the text data independently of the audio, $99.99\%$ efficiency could be achieved with a speed-up of $2.24x$.

---

[4]https://github.com/mlcommons/training/tree/master/rnn_speech_recognition/pytorch

# N    Packing Paper Abstracts (PubMed)

This section analyses the length of abstracts to give an intuition about how different documents can be in length. Figure 14 depicts the length of abstracts in characters extracted from PubMed.[5] If these abstracts were directly used as sequences, a character length of 1000 could result in $1.9x$ speed-up from packing. The potential speed-ups for length 2000, 3000, 4000 would be $2x$, $3x$, and $4x$, respectively. Note that, document clean-up procedures would usually eliminate documents that are too short or too long for data sanitizing purposes.

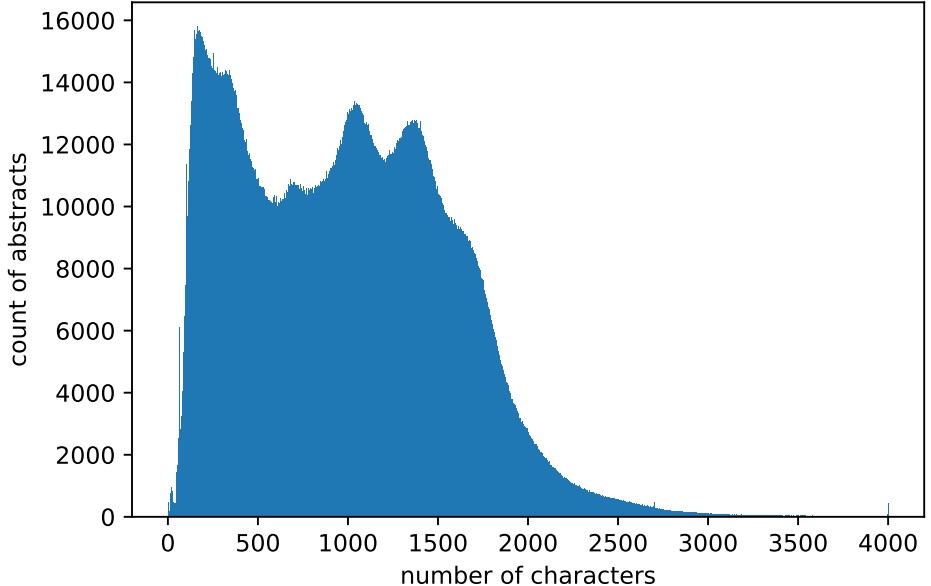

Figure 14: Abstract length distribution in PubMed.

Note that for the processing in BlueBERT [45], paper titles and abstracts get separated into sequences, tokenized, and then combined with the BERT sequence combination approach for a maximum sequence length of 128 tokens. Thus, it results in a different distribution.

---

[5]https://huggingface.co/datasets/pubmed

# O   MLPerf™ phase 2 learning curves

1100   This section provides further learning curves related to Section 4.2.

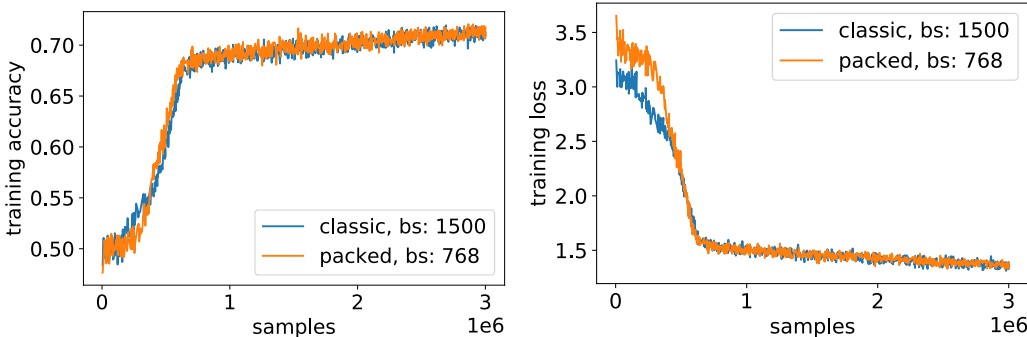

Figure 15: Comparison of learning curves for packed and unpacked processing with **reduced batch size** for the packed approach.

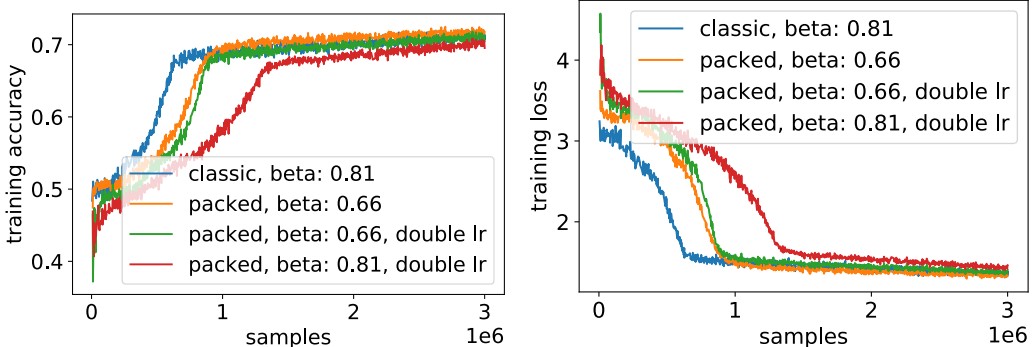

Figure 16: Comparison of learning curves for packed and unpacked processing with **heuristics** applied.

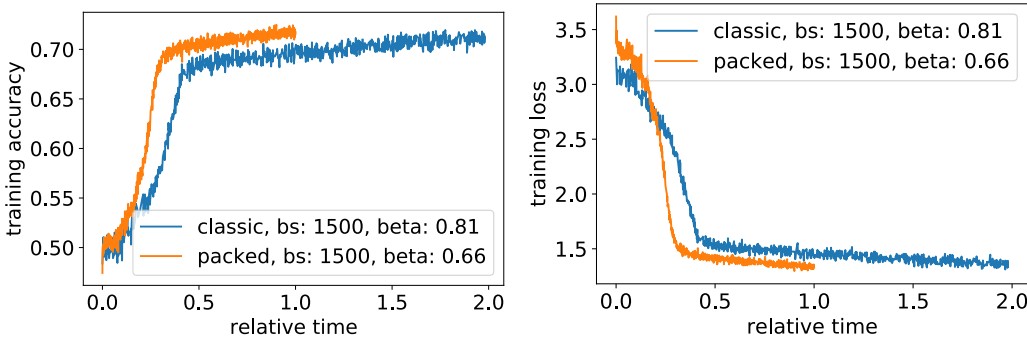

Figure 17: Comparison of learning curves for packed and unpacked processing in the **optimized setup**.

 # P    Full pretraining of BERT base and large learning curves

 This section provides further learning curves related to Section 4.3.

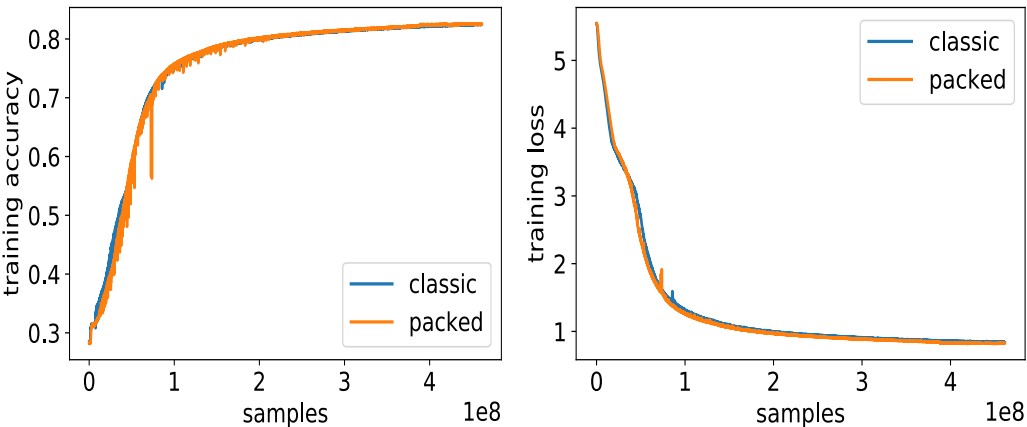

Figure 18: Comparison of learning curves for **BERT base phase 1** (sequence length 128) with packed and unpacked processing.

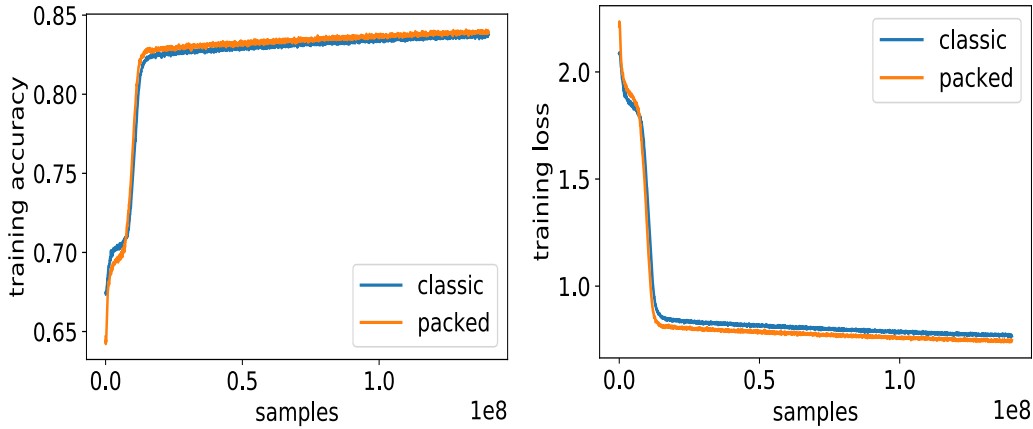

Figure 19: Comparison of learning curves for **BERT base phase 2** (sequence length 384) with packed and unpacked processing.

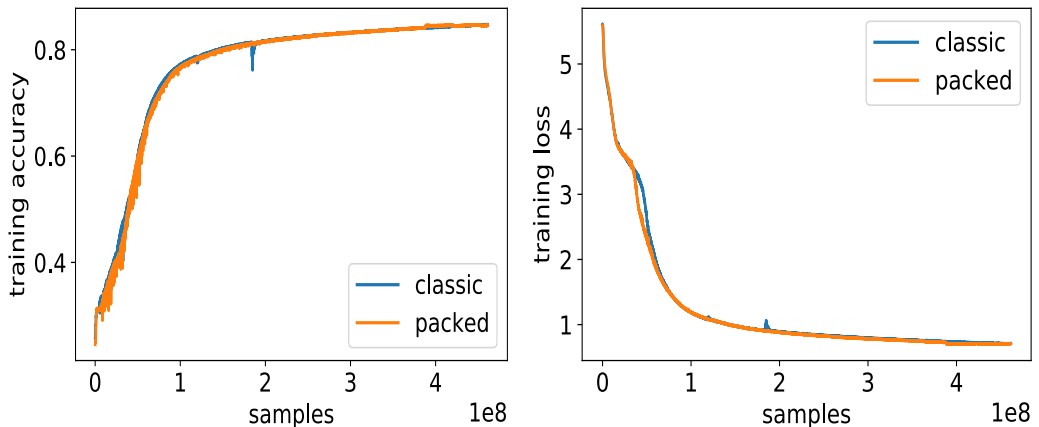

Figure 20: Comparison of learning curves for **BERT large phase 1** (sequence length 128) with packed and unpacked processing.

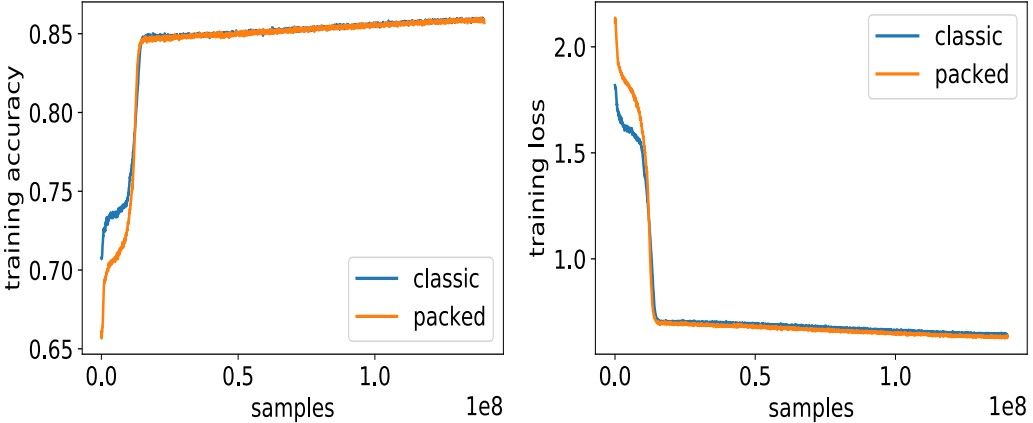

Figure 21: Comparison of learning curves for **BERT large phase 2** (sequence length 384) with packed and unpacked processing.

## Q    Note on changing the sequence length for optimal packing

An interesting aspect of packing is that the maximum sequence length for packing could be larger than the maximum sequence length in the underlying dataset that gets packed.

For the QM9 dataset, this means that by setting the maximum sequence length to 36 instead of 27 an optimal $1.6x$ speed-up can be easily achieved.

Note that the choice of maximum sequence length depends on the underlying machine learning algorithm. Due to the squared computational and memory complexity of self-attention in BERT and other transformers, the maximum sequence length is usually kept as small as possible for these models. So an increase for packing alone is not practical. For algorithms with linear complexity as for example Graph Neural Networks, implemented in PyG, larger maximum sequence length can be chosen to ensure, optimal packing is always possible.

## R    Fine-tuned longest-pack-first histogram-packing

In the main paper, we focused on SPFHP due its simplicity. In this section, we analyse the effect of applying the "Best-Fit" algorithm [12]. Here, the longest pack that still fits the sequence is chosen instead of the shortest one. In contrast to SPFHP, we additionally consider splitting the histogram count, if it can fit multiple times. A simple example is sequence length 256, where we divide the respective histogram count by 2 to create the optimal pack with strategy $[256, 256]$ instead of the strategy $[256]$. This latter strategy would be complemented by other sequences but would probably not result in an optimal packing. The implementation of this approach is much more complex than the SPFHP implementation. The code is provided in Listing 8 and the results in Table 7.

| pack. depth | # strat. used | # packs | # tokens | # padding tokens | efficiency (%) | pack. factor |
|---|---|---|---|---|---|---|
| 1 | 508 | 16279552 | 8335130624 | 4170334451 | 49.967 | 1.000 |
| 2 | 634 | 10099081 | 5170729472 | 1005933299 | 80.546 | 1.612 |
| 3 | 648 | 9090154 | 4654158848 | 489362675 | 89.485 | 1.791 |
| 4 | 671 | 8657119 | 4432444928 | 267648755 | 93.962 | 1.880 |
| 8 | 670 | 8207569 | 4202275328 | 37479155 | 99.108 | 1.983 |
| 16 | 670 | 8140006 | 4167683072 | 2886899 | 99.931 | 2.000 |
| 29/max | 670 | 8138483 | 4166903296 | 2107123 | 99.949 | 2.000 |

Table 7: Performance results of longest-pack-first histogram-packing for Wikipedia BERT pre-training with maximum sequence length $512$.

We can see that longest-pack-first histogram-packing (LPFHP) uses a much higher packing depth when no limit is set (29 instead of 16). Splitting the histogram counts results in slightly higher numbers of used strategies compared to SPFHP where the number of used strategies is limited by the maximum sequence length. The best efficiency of LPFHP is $99.949\%$ with packing factor of 2 which is slightly higher than the $99.75\%$ (1.996 packing factor) for NNLSHP and $99.6\%$ for SPFHP (1.993 packing factor). All algorithms are very close to the upper limit.

Note that for NNLSHP, we only fill up the unpacked samples with padding. Applying best-fit on the remains, similar results can be expected. Although the benefits of the improved algorithm are negligible, we share the concept and code below in case packing is applied to other data with a different distribution that would benefit more from it, or for applications where only perfectly packed sequences without padding are of interest.

## S  Extended NNLS with padding token weighting

In Section F.4.4, we defined the residual as

$$r = b - A \cdot round(x) \qquad (21)$$

and discovered that a positive residual corresponds to sequences that we did not pack at all and should be avoided. Negative residuals correspond to padding and should be minimized. Due to this discrepancy, we decided to set small weights for very short sequences (that don't occur in the data). However, it was not possible to directly optimize the amount of padding. A negative residual component for length $i$, $r_i$, results in $|r_i| \cdot i$ padding tokens, however a positive residual actually results into $(512 - r_i) \cdot i$ padding tokens. This cannot be addressed by our weighting approach in

$$\min_{x \in \mathbb{R}^m} \quad \|(wA) \cdot x - (wb)\|^2$$
$$\text{s.t.} \quad x \geq 0 \qquad (22)$$

Working within the NNLS approach, we can strictly enforce a non-positive residual $r$ (before rounding to integer). To that end, we define a new auxiliary variable $\overline{r} \approx -(b - Ax)$ which is the negative of the residual, $r$. This will allow us to reformulate the objective $r \leq 0$ to the non-negative constraint: $\overline{r} \geq 0$.

$$\min_{x \in \mathbb{R}^m} \quad \|(wA) \cdot x - (wb)\|^2 + \|\overline{w} \cdot A \cdot x - \overline{w} \cdot b - \overline{w} \cdot \overline{r}\|^2$$
$$\text{s.t.} \quad x \geq 0 \qquad (23)$$
$$\overline{r} \geq 0$$

This will enforce $\overline{r} = Ax - b \geq 0$ due to the large weight, $\overline{w} := 10^6$, and no upper limits on $\overline{r}$. Now, we can set $w_i := i$ to optimize for the padding tokens. Due to the use of the squared error, we would however optimize the squared sum of padding tokens instead of the preferred sum of padding tokens. To accomplish the latter, we would have to replace the L2-norm problem by an L1-norm problem which would be too complex to solve. Note that due to rounding, the unwanted positive residuals $r$ ($\overline{r} < 0$) might still occur. This could be avoided by rounding up $x$ instead of normal rounding of $x$. To put the new formulation into a solver, we replace

$$b \text{ by } \begin{pmatrix} b \\ b \end{pmatrix}, \ x \text{ by } \begin{pmatrix} x \\ \overline{r} \end{pmatrix}, \ w \text{ by } \begin{pmatrix} w \\ \overline{w} \end{pmatrix}, \text{ and } A \text{ by } \begin{pmatrix} A & 0_m \\ A & -D_m \end{pmatrix}, \qquad (24)$$

where $0_m$ is an $m \times m$ matrix with $m$ being the maximum sequence length, 512, and $D_m$ is a unit matrix of the same dimensions as $0_m$. Since, we are already close to optimum especially on the Wikipedia dataset, the results are only a little bit better. The processing time however increases from 30 to 415 seconds without considering the increased time for constructing the processing matrix. Since the slightly improved algorithm might be nevertheless relevant for other applications, we share it in Listing 9.

 **T   Packing source code**

Listing 2: Non-negative least squares histogram-packing

```python
import time
import numpy as np
from scipy import optimize, stats
from functools import lru_cache

def get_packing_matrix(strategy_set, max_sequence_length):
    num_strategies = len(strategy_set)
    A = np.zeros((max_sequence_length, num_strategies), dtype=np.int32)
    for i, strategy in enumerate(strategy_set):
        for seq_len in strategy:
            A[seq_len - 1, i] += 1
    return A

@lru_cache(maxsize=None)
def get_packing_strategies(start_length, minimum_increment, target_length, depth):
    gap = target_length - start_length
    strategies = []
    # Complete the packing with exactly 1 number
    if depth == 1:
        if gap >= minimum_increment:
            strategies.append([gap])
    # Complete the sample in "depth" steps, recursively
    else:
        for new in range(minimum_increment, gap + 1):
            new_gap = target_length - start_length - new
            if new_gap == 0:
                strategies.append([new])
            else:
                options = get_packing_strategies(start_length + new, new, target_length, depth - 1)
                for option in options:
                    if len(option) > 0:
                        strategies.append([new] + option)
    return strategies

def pack_using_nnlshp(histogram, max_sequence_length, max_sequences_per_pack):
    # List all unique ways of packing to the desired maximum sequence length
    strategy_set = get_packing_strategies(0, 1, max_sequence_length, max_sequences_per_pack)
    print(f"Packing will involve {len(strategy_set)} unique packing strategies.")
    # Get the packing matrix corresponding to this list of packing strategies
    A = get_packing_matrix(strategy_set, max_sequence_length)
    # Weights that penalize the residual on short sequences less.
    penalization_cutoff = 8
    w0 = np.ones([max_sequence_length])
    w0[:penalization_cutoff] = 0.09
    # Solve the packing problem
    print(f"Sequences to pack: ", histogram.sum())
    start = time.time()
    strategy_repeat_count, rnorm = optimize.nnls(np.expand_dims(w0, -1) * A, w0 * histogram)
    print(f"Solving non-negative least squares took {time.time() - start:3.2f} seconds.")
    # Round the floating point solution to nearest integer
    strategy_repeat_count = np.rint(strategy_repeat_count).astype(np.int64)
    # Compute the residuals, shape: [max_sequence_length]
    residual = histogram - A @ strategy_repeat_count
    # Handle the left-over sequences i.e. positive part of residual
    unpacked_seqlen = np.arange(1, max_sequence_length + 1)[residual > 0]
    for l in unpacked_seqlen:
        strategy = sorted([l, max_sequence_length - l])  # the depth 1 strategy
        strategy_index = strategy_set.index(strategy)
        strategy_repeat_count[strategy_index] += residual[l-1]
    # Re-compute the residual with the updated strategy_repeat_count
    # This should now be strictly < 0
    residual = histogram - A @ strategy_repeat_count
    # Add padding based on deficit (negative residual portion of residual)
    padding = np.where(residual < 0, -residual, 0)
    # Calculate some basic statistics
    sequence_lengths = np.arange(1, max_sequence_length + 1)
    old_number_of_samples = histogram.sum()
    new_number_of_samples = int(strategy_repeat_count.sum())
    speedup_upper_bound = 1.0/(1 - (histogram*(1 - sequence_lengths / max_sequence_length)).sum()/old_number_of_samples)
    num_padding_tokens_packed = (sequence_lengths * padding).sum()
    efficiency = 1 - num_padding_tokens_packed/(new_number_of_samples*max_sequence_length)
    print(f"Packing efficiency (fraction of real tokens): {efficiency:3.4f}\n",
          f"Speed-up theoretical limit: {speedup_upper_bound:3.4f}\n",
          f"Achieved speed-up over un-packed dataset: {old_number_of_samples/new_number_of_samples:3.5f}")
    return strategy_set, strategy_repeat_count
```

## Listing 3: Shortest-pack-first histogram-packing

```python
from collections import defaultdict
import numpy as np

def add_pack(pack, count, tmp, final, limit, offset):
    """Filter out packs that reached maximum length or number of sequences."""
    if len(pack) == limit or offset == 0:
        final[offset].append((count, pack))
    else:
        tmp[offset].append((count, pack))

def pack_using_spfhp(histogram, max_sequence_length, max_sequences_per_pack):
    """Shortest-pack-first histogram-packing algorithm."""
    reversed_histogram = np.flip(histogram)
    # Initialize main strategy data dictionary.
    # The key indicates how many tokens are left for full length.
    # The value is a list of tuples, consisting of counts and respective packs.
    # A pack is a (sorted) list of sequence length values that get concatenated.
    tmp_strategies_per_length = defaultdict(list)
    strategies_per_length = defaultdict(list)
    # Index i indicates here, how much space is left, due to reversed histogram
    for i in range(max_sequence_length):
        n_sequences_to_bin = reversed_histogram[i]
        length_to_bin = max_sequence_length - i
        offset = i + 1  # largest possible offset
        while n_sequences_to_bin > 0:
            if (length_to_bin + offset) in tmp_strategies_per_length:
                # extract shortest pack that will get modified
                n_sequences_to_pack, pack = tmp_strategies_per_length[
                    length_to_bin + offset].pop()
                new_pack = pack + [length_to_bin]
                count = min(n_sequences_to_pack, n_sequences_to_bin)
                if n_sequences_to_pack > n_sequences_to_bin:
                    # old pack gets reduced
                    n_sequences_to_pack -= n_sequences_to_bin
                    tmp_strategies_per_length[length_to_bin + offset].append(
                        (n_sequences_to_pack, pack))
                    n_sequences_to_bin = 0
                else:
                    n_sequences_to_bin -= n_sequences_to_pack
                add_pack(new_pack, count,
                         tmp_strategies_per_length, strategies_per_length,
                         max_sequences_per_pack, offset)
                # clean up to speed up main key search
                if not tmp_strategies_per_length[length_to_bin + offset]:
                    tmp_strategies_per_length.pop(length_to_bin + offset)
            else:
                offset -= 1
            # Does not fit anywhere. Create new pack.
            if offset < 0:
                add_pack([length_to_bin], n_sequences_to_bin,
                         tmp_strategies_per_length, strategies_per_length,
                         max_sequences_per_pack, i)
                n_sequences_to_bin = 0
    # merge all strategies
    for key in tmp_strategies_per_length:
        strategies_per_length[key].extend(tmp_strategies_per_length[key])
    # flatten strategies dictionary
    strategy_set = []
    strategy_repeat_count = []
    for key in strategies_per_length:
        for count, pack in strategies_per_length[key]:
            pack.reverse()
            strategy_set.append(pack)
            strategy_repeat_count.append(count)
    return strategy_set, np.array(strategy_repeat_count)
```

## Listing 4: Evaluation function of shortest-pack-first histogram-packing

```python
"""Max depth analysis of shortest-pack-first histogram-packing."""
from collections import defaultdict
import tabulate
import time
import numpy as np

def evaluate_spfhp(histogram, max_sequence_length):
    """Evaluate shortest-pack-first histogram-packing algorithm."""
    stats_data = [["pack. depth", "# strat. used", "# packs", "# tokens",
                   "# padding tok.", "efficiency (%)", "pack.factor", "time"]]
    for max_sequences_per_pack in [1, 2, 3, 4, 8, 16, "max"]:
        start = time.time()
        strategy_set, strategy_repeat_count = pack_using_spfhp(
            histogram, max_sequence_length, max_sequences_per_pack)
        duration = time.time() - start

        # Performance Evaluation of packing approach
        n_strategies = int(len(strategy_set))
        packs = int(sum(strategy_repeat_count))
        sequences = sum([count*len(pack) for count, pack in
                         zip(strategy_repeat_count, strategy_set)])
        total_tokens = int(max_sequence_length * packs)
        empty_tokens = int(sum([
            count*(max_sequence_length-sum(pack)) for count, pack in
            zip(strategy_repeat_count, strategy_set)]))
        token_efficiency = 100 - empty_tokens / total_tokens * 100
        if max_sequences_per_pack == "max":
            m_length = max([len(pack) for pack in strategy_set])
            max_sequences_per_pack = "max ({})".format(m_length)
        stats_data.append([
            max_sequences_per_pack, n_strategies, packs, total_tokens,
            empty_tokens, token_efficiency, sequences / packs, duration])
    print(tabulate.tabulate(stats_data, headers="firstrow", floatfmt=".3f"))
```

## Listing 5: Loss calculation

```python
# The number of sequences in each batch may vary
sequences_in_batch = tf.reduce_sum(tf.reduce_max(masked_lm_weight, -1))
sequences_in_batch = tf.cast(sequences_in_batch, tf.float32)
# Create the 0/1 mask that will be used to un-packed sequences
masked_lm_weight = tf.reshape(masked_lm_weight, [B, 1, -1])
sequence_selection = tf.reshape(tf.range(1, max_sequences_per_pack + 1), [1, -1, 1])
sequence_selection = tf.cast(masked_lm_weight == sequence_selection, tf.float32)
# Apply the mask to un-pack the loss per sequence
nll_per_token = tf.reshape(nll_per_token, [B, 1, -1])
nll_per_sequence = sequence_selection * nll_per_token
# Normalize the per-sequence loss by the number of mlm-tokens in the sequence (as is standard)
attempted = tf.reduce_sum(sequence_selection, -1, keepdims=True)
attempted = attempted + tf.cast(attempted == 0, tf.float32)  # prevent NaNs when dividing by attempted
nll_per_sequence = nll_per_sequence/attempted
# Average per-batch loss (so contributions from different batches are comparable)
lm_loss = tf.reduce_sum(nll_per_sequence)/sequences_in_batch
```

## Listing 6: Wikipedia and SQuAD 1.1 histograms

```python
"""Wikipedia and SQUaD 1.1 histograms.

For sequence length 128 to 512, we use the Wikipedia article dump from October 1st 2020.
For sequence length 1024 and 2048, we use the Wikipedia article dump from February 8th 2021.
Duplication factors slightly differ.
"""
import numpy as np
wikipedia_histogram = np.array([
    0, 0, 0, 0, 1821, 1226, 1969, 1315, 1794, 1953, 3082, 3446, 4166, 5062,
    9554, 16475, 19173, 17589, 17957, 19060, 21555, 23524, 26954, 30661, 33470, 36614, 40134, 43256,
    46094, 49350, 52153, 55428, 58109, 60624, 63263, 64527, 65421, 66983, 68123, 68830, 70230, 70486,
    72467, 72954, 73955, 74311, 74836, 74489, 74990, 75377, 74954, 75096, 74784, 74698, 74337, 74638,
    74370, 73537, 73597, 73153, 72358, 71580, 71082, 70085, 69733, 69445, 67818, 67177, 66641, 65709,
    64698, 63841, 63218, 62799, 61458, 60848, 60148, 59858, 58809, 58023, 56920, 55999, 55245, 55051,
    53979, 53689, 52819, 52162, 51752, 51172, 50469, 49907, 49201, 49060, 47948, 47724, 46990, 46544,
    46011, 45269, 44792, 44332, 43878, 43984, 42968, 42365, 42391, 42219, 41668, 41072, 40616, 40587,
    39999, 40169, 39340, 38906, 38438, 38142, 37757, 37818, 37535, 37217, 36757, 36589, 36151, 35953,
    35531, 35496, 35089, 35053, 34567, 34789, 34009, 33952, 33753, 33656, 33227, 32954, 32686, 32880,
    32709, 31886, 32126, 31657, 31466, 31142, 31106, 30650, 30316, 30494, 30328, 30157, 29611, 29754,
    29445, 28921, 29271, 29078, 28934, 28764, 28445, 28319, 28141, 28282, 27779, 27522, 27333, 27470,
    27289, 27102, 27018, 27066, 26925, 26384, 26188, 26385, 26392, 26082, 26062, 25660, 25682, 25547,
    25425, 25072, 25079, 25346, 24659, 24702, 24862, 24479, 24288, 24127, 24268, 24097, 23798, 23878,
    23893, 23817, 23398, 23382, 23280, 22993, 23018, 23242, 22997, 22894, 22470, 22612, 22452, 21996,
    21843, 22094, 21916, 21756, 21955, 21444, 21436, 21484, 21528, 21597, 21301, 21197, 21281, 21066,
    20933, 21023, 20888, 20575, 20574, 20511, 20419, 20312, 20174, 20023, 20087, 19955, 19946, 19846,
    19562, 19710, 19556, 19477, 19487, 19387, 19225, 19069, 19360, 18655, 19034, 18763, 18800, 19012,
    18893, 18714, 18645, 18577, 18317, 18458, 18374, 18152, 17822, 18102, 17735, 17940, 17805, 17711,
    17690, 17703, 17669, 17410, 17583, 17331, 17313, 16892, 16967, 16870, 16926, 17233, 16845, 16861,
    16576, 16685, 16455, 16687, 16747, 16524, 16473, 16349, 16273, 16255, 16228, 16219, 16021, 16111,
    15867, 15751, 16081, 15703, 15751, 15854, 15665, 15469, 15431, 15428, 15464, 15517, 15335, 15461,
    15237, 15292, 15305, 15351, 15078, 14810, 15119, 14780, 14664, 14869, 14722, 14890, 14672, 14439,
    14685, 14706, 14840, 14373, 14286, 14596, 14615, 14418, 14299, 13987, 14167, 14107, 14096, 14202,
    13985, 14118, 14094, 14127, 13896, 13864, 13597, 13572, 13717, 13669, 13782, 13617, 13284, 13333,
    13425, 13457, 13256, 13404, 13318, 13425, 13317, 13179, 13193, 13257, 13160, 12813, 13149, 13010,
    12867, 12958, 12818, 12801, 12749, 12810, 12575, 12673, 12514, 12735, 12523, 12677, 12298, 12469,
    12341, 12445, 12477, 12326, 12110, 12087, 12305, 12156, 12032, 12190, 12150, 11980, 12022, 11825,
    11969, 11831, 11997, 11924, 11739, 11685, 11702, 11783, 11783, 11659, 11647, 11610, 11526, 11577,
    11538, 11536, 11497, 11480, 11374, 11234, 11433, 11466, 11475, 11147, 11376, 11217, 11002, 11245,
    11124, 11000, 11129, 10923, 10966, 11071, 11029, 11036, 10972, 11012, 10800, 10936, 10904, 10750,
    10669, 10766, 10780, 10675, 10905, 10511, 10598, 10583, 10658, 10471, 10667, 10601, 10430, 10440,
    10510, 10148, 10468, 10346, 10257, 10286, 10235, 10351, 10182, 10095, 10192, 9866, 10070,
    10148, 9956, 10132, 10043, 9741, 10003, 10056, 9920, 10021, 9838, 9854, 9740, 9782, 9799,
    9798, 9788, 9840, 9747, 9797, 9893, 9593, 9535, 9658, 9554, 9593, 9530, 9523, 9488,
    9548, 9418, 9418, 9508, 9638, 9521, 9277, 9289, 9255, 9322, 9281, 9351, 9259, 9255,
    9225, 9098, 9268, 9227, 9224, 9106, 9239, 3815044], dtype=np.int64)

wikipedia_max_sequence_length = 512

wikipedia_128_histogram = np.array([
    0, 0, 0, 0, 3101, 1980, 3129, 1999, 2921, 3125, 4830, 5364, 6732, 8047,
    13409, 21166, 25207, 25106, 27446, 30336, 35090, 39592, 45885, 52030, 57859, 64301, 71861, 78013,
    84925, 91873, 98489, 104534, 112174, 117841, 124085, 129462, 133240, 138870, 143228, 146717, 151324,
    154822, 158681, 162508, 165513, 168386, 170678, 172157, 174582, 174811, 177932, 177775, 179075, 178718,
    179454, 179142, 179395, 178585, 178799, 177238, 176319, 174648, 173217, 174185, 172356, 170476, 168799,
    166638, 166251, 163258, 161835, 160796, 158675, 157306, 156076, 154365, 153016, 151754, 150507, 148666,
    146567, 144652, 143753, 141893, 140452, 139608, 138186, 136564, 135683, 134562, 132625, 132270, 129838,
    130280, 128484, 127725, 126559, 125192, 124847, 124314, 123023, 122125, 121434, 120822, 119386, 119410,
    117987, 118109, 116432, 116579, 114937, 114728, 114064, 114111, 113091, 112457, 111797, 111032, 111055,
    109929, 110613, 109024, 109634, 109102, 108301, 107099, 106661, 21454463], dtype=np.int64)

wikipedia_128_max_sequence_length = 128

wikipedia_384_histogram = np.array([
    0, 0, 0, 0, 1996, 1380, 2227, 1385, 1908, 2065, 3221, 3673, 4581, 5391,
    9975, 16932, 19431, 18385, 19107, 20129, 23118, 24966, 29088, 35695, 38943, 43618, 46724,
    50553, 53774, 57470, 60695, 63903, 67021, 69559, 71609, 72274, 73630, 75620, 76946, 78870, 79774,
    81019, 82236, 83350, 84128, 84939, 84585, 85703, 85151, 85245, 85923, 85869, 85748, 85704, 85459,
    84822, 84487, 83940, 84322, 82652, 82371, 81509, 80958, 80255, 79266, 77896, 76827, 76356, 75703,
    74378, 73639, 72827, 71460, 70859, 69590, 69009, 67987, 66779, 65626, 65372, 63939, 63290, 62662,
    61334, 61194, 60371, 59318, 58753, 57841, 57492, 56965, 55816, 55709, 54678, 54572, 53805, 53126,
    52578, 51656, 51337, 50926, 50590, 50018, 49860, 48821, 48788, 48365, 47776, 47225, 46417, 46438,
    45922, 45626, 45021, 44818, 44293, 44338, 43474, 43547, 42987, 42685, 42425, 42256, 41729, 41583,
    41194, 40717, 40565, 40238, 39761, 39557, 39285, 39009, 38955, 38841, 38212, 37846, 37808, 37609,
    37852, 37513, 36960, 36903, 36265, 36026, 36135, 35781, 35531, 35381, 34939, 35241, 34523, 34547,
    34106, 34106, 33687, 34008, 33531, 33630, 33335, 32980, 32756, 32666, 32421, 32135, 32290, 32395,
    31661, 31958, 31580, 31290, 31074, 31199, 30740, 30577, 30244, 30305, 30238, 30171, 29987, 29783,
    29765, 29162, 29584, 29470, 29137, 29254, 29018, 28646, 28788, 28470, 28295, 28465, 28114, 28241,
    28001, 27736, 27501, 27677, 27724, 27415, 27378, 27397, 27194, 26876, 26929, 26597, 26475, 26326,
    26278, 26246, 25962, 25901, 25916, 25540, 25514, 25701, 25954, 25284, 25452, 24888, 25051, 24975,
    24900, 24736, 24554, 24605, 24558, 24828, 24273, 23974, 24305, 24229, 23824, 24006, 23606, 23748,
    23496, 23262, 23477, 23510, 23089, 23185, 23289, 22947, 22999, 22879, 22846, 22564, 22942, 22512,
    22245, 22468, 22453, 22454, 22073, 22081, 21918, 21799, 21721, 21641, 21994, 21542, 21441, 21438,
    21634, 21360, 21237, 21327, 20946, 20841, 20701, 21044, 20797, 20810, 20758, 20616, 20717,
    20370, 20444, 20365, 20420, 20263, 20046, 19942, 20301, 20086, 19971, 19798, 19579, 19720, 19676,
    19526, 19330, 19325, 19385, 19095, 19333, 19286, 18955, 19190, 19149, 18929, 18867, 18912, 18954,
    18975, 18773, 18808, 18896, 18648, 18540, 18461, 18551, 18367, 18474, 18366, 18407, 18304, 18071,
    18276, 18302, 18367, 18223, 18077, 17848, 18055, 17895, 17757, 17755, 17534, 17617, 17292, 17452,
    17367, 17484, 17480, 17456, 17212, 17454, 17548, 17296, 17000, 17289, 17032, 17151, 17113, 16942,
    16955, 16744, 16922, 17037, 16971, 16736, 16945, 16637, 16703, 16328, 16587, 16339, 16404, 16492,
    16525, 16309, 16374, 16262, 16180, 16202, 16021, 16042, 16129, 16101, 15986, 16197, 15792, 15935,
    15914, 15915, 15902, 15688, 15717, 5676254]
```

```python
, dtype=np.int64)

wikipedia_384_max_sequence_length = 384

wikipedia_1024_histogram = np.array([
    0, 0, 0, 0, 7363, 4744, 8434, 5610, 13205, 6932, 10664, 13887,
    16118, 24347, 31871, 66246, 77082, 65887, 66852, 69969, 79068, 86941, 99807, 111153,
    123160, 137381, 154228, 166304, 180331, 192040, 206214, 215316, 227387, 238863, 247444, 253057,
    258237, 262474, 266124, 269895, 275211, 277955, 280852, 283614, 286648, 287714, 291932, 292063,
    292252, 292122, 291963, 291950, 290741, 289930, 289635, 288843, 289106, 285626, 283735, 283763,
    279961, 277485, 275528, 274559, 271725, 269530, 266926, 263998, 262027, 259506, 256167, 253231,
    251842, 249295, 246119, 243579, 240920, 239550, 236008, 232477, 228900, 226724, 222639, 220947,
    217754, 215699, 213277, 209415, 209497, 206063, 202650, 201057, 199017, 196767, 194504, 192778,
    190108, 188113, 186489, 184212, 182828, 181271, 179863, 177707, 174891, 173822, 172668, 171383,
    168696, 167579, 165974, 164577, 163931, 161678, 160632, 158468, 157537, 155880, 154696, 154374,
    152753, 151583, 150617, 149261, 148185, 146336, 145928, 143589, 142916, 141994, 140233, 140480,
    139865, 138102, 137013, 136298, 135120, 133563, 133063, 131795, 131001, 130944, 129157, 128813,
    127434, 127698, 126006, 124766, 123580, 123936, 122788, 121985, 121212, 119757, 118557, 118198,
    117536, 117253, 116175, 116240, 115372, 114303, 113935, 113271, 112221, 111883, 110628, 110057,
    109411, 109347, 108960, 108049, 107465, 106268, 105262, 105826, 105049, 103570, 104051, 103013,
    101732, 101998, 101922, 100885, 100328, 99803, 99771, 99120, 98958, 98036, 97766, 97099,
    95960, 95916, 94781, 94124, 94467, 93805, 92947, 93067, 92161, 91783, 91722, 91620,
    90588, 90104, 89736, 89196, 88915, 88424, 87636, 87356, 87247, 86421, 86743, 86135,
    85400, 85421, 84616, 84760, 84117, 84004, 83306, 82563, 82220, 81649, 81791, 81767,
    81101, 80423, 80860, 79756, 79404, 78844, 78655, 78712, 77841, 77453, 77561, 76647,
    76480, 76123, 76217, 76223, 76105, 75057, 74794, 74204, 73918, 74153, 74136, 73317,
    73022, 72178, 71935, 71819, 71835, 70887, 70521, 70501, 69927, 70242, 70127, 68686,
    69069, 68544, 68655, 68127, 68341, 67440, 67554, 67010, 66569, 66745, 66429, 66271,
    65694, 65858, 64893, 64461, 64710, 64451, 64060, 64068, 63082, 63415, 63325, 62978,
    63069, 62079, 62130, 62529, 61961, 61093, 61260, 60825, 60348, 60187, 60726, 60106,
    59547, 59172, 60090, 59104, 58742, 58683, 58425, 58537, 58229, 57599, 57673, 57604,
    57433, 56886, 56289, 56343, 56168, 56058, 56437, 55851, 55882, 55346, 55218, 55496,
    55359, 54481, 54448, 54634, 54026, 53694, 54213, 53115, 53392, 53114, 53451, 52686,
    51918, 52538, 52225, 51882, 51453, 51946, 51433, 51036, 51706, 51381, 51154, 50810,
    50705, 50615, 49501, 49823, 49730, 49855, 49268, 49119, 48979, 48909, 48687, 48603,
    48227, 47873, 48152, 48029, 48530, 47844, 47209, 47368, 46891, 46944, 46450, 46501,
    46729, 46052, 46148, 45931, 46702, 46161, 45322, 45557, 45583, 45433, 45154, 44824,
    44827, 44354, 44175, 44192, 44053, 43849, 43935, 43927, 43549, 43493, 43250, 43172,
    42918, 42648, 42747, 42936, 42206, 42169, 41825, 42190, 41973, 42001, 41717, 41141,
    41118, 41419, 41234, 41084, 41170, 41027, 40836, 40740, 40454, 40242, 40343, 39910,
    39512, 39971, 39321, 39238, 39143, 39453, 39048, 38997, 38995, 38984, 38588, 39064,
    38165, 38726, 38215, 37930, 37995, 37974, 38212, 37397, 37367, 37573, 37331, 37215,
    36850, 36864, 36801, 36822, 36686, 36479, 36390, 36341, 36355, 35850, 36282, 35294,
    35433, 35698, 35534, 35105, 35066, 35092, 34855, 35046, 34559, 34548, 34376, 34918,
    34782, 34416, 34643, 34643, 34022, 34078, 33797, 33601, 33636, 33455, 33513, 33516,
    33222, 33694, 33371, 32986, 33058, 32760, 32795, 32638, 33060, 32696, 32659, 32522,
    32400, 32230, 31852, 31913, 32168, 31532, 31490, 31728, 31518, 31333, 31496, 31117,
    31206, 31317, 31273, 30896, 30977, 31021, 30815, 30858, 30618, 30313, 30219, 30504,
    30113, 30292, 30073, 30073, 29820, 29749, 29319, 29727, 29824, 29448, 29068, 29252,
    28837, 29217, 29361, 28997, 28648, 29087, 29048, 28700, 28716, 28636, 28346, 28442,
    28575, 28541, 28255, 28145, 27853, 28094, 27706, 27422, 28158, 27347, 27292, 27993,
    27487, 27375, 27503, 27508, 27200, 27160, 27336, 26888, 26960, 26876, 26422, 26896,
    26592, 26752, 26713, 26290, 26289, 26379, 26003, 26044, 26407, 25659, 26243, 25573,
    25477, 25590, 25717, 25333, 25555, 25537, 25303, 25326, 25035, 25290, 25129, 25184,
    24704, 24886, 24818, 24895, 24793, 24598, 24644, 24837, 24761, 24576, 24419, 24304,
    24285, 23889, 24080, 23894, 23900, 23916, 23891, 23838, 23704, 23632, 23503, 23316,
    23646, 23490, 23438, 23541, 22810, 23053, 23151, 22921, 22966, 23220, 22938, 22880,
    22871, 23104, 22819, 22737, 22806, 22293, 22722, 22652, 22288, 22068, 22119, 22244,
    21987, 22228, 21901, 21529, 21973, 21807, 21748, 21729, 21713, 21548, 21501, 21695,
    21691, 21408, 21589, 21341, 21576, 21349, 21247, 21217, 21294, 21083, 21479, 21414,
    21021, 21200, 21057, 20713, 20708, 20994, 20569, 20643, 20621, 20649, 20672, 20438,
    20550, 20299, 20323, 20269, 20529, 20150, 20371, 20306, 20331, 20453, 20064, 20243,
    20080, 20010, 20082, 19786, 19631, 19588, 19450, 19764, 19690, 19757, 19768, 19456,
    19312, 19364, 19347, 19194, 19244, 19027, 19303, 19117, 19070, 19019, 18888, 18706,
    18802, 18690, 18827, 18507, 18431, 18523, 18582, 18389, 18624, 18446, 18506, 18615,
    18559, 18049, 18322, 18004, 18211, 18341, 18348, 18462, 17997, 18105, 18038, 17843,
    17788, 18096, 17998, 18100, 17634, 17881, 17808, 17655, 17622, 17589, 17609, 17403,
    17727, 17569, 17443, 17382, 17526, 17521, 17602, 17079, 17547, 17027, 17338, 17052,
    17674, 16956, 17100, 16919, 17032, 16887, 16924, 16730, 16828, 16828, 16831, 16926,
    16588, 16463, 16655, 16723, 16658, 16414, 16808, 16506, 16465, 16579, 16287, 16365,
    16158, 16268, 16330, 16304, 16578, 16288, 16207, 16257, 16007, 15787, 15981, 15994,
    15842, 15995, 15946, 15877, 15682, 15788, 15691, 15981, 15714, 15521, 15576, 15716,
    15573, 15558, 15673, 15422, 15266, 15369, 15288, 15612, 15327, 15325, 15182, 15177,
    15186, 15257, 15354, 15283, 15152, 15220, 14798, 14938, 15041, 14849, 15315, 14860,
    14903, 14759, 14883, 14678, 14862, 14816, 14581, 14905, 14843, 14595, 14903, 14687,
    14437, 14416, 14561, 14263, 14321, 14534, 14571, 14353, 14188, 14097, 14306, 14413,
    14141, 14363, 14199, 14102, 14091, 14263, 14145, 14080, 14058, 13890, 14070, 13861,
    14216, 13963, 13852, 13952, 13890, 13679, 13932, 13856, 13672, 13723, 13660, 13822,
    13891, 13699, 13534, 13495, 13875, 13617, 13649, 13567, 13585, 13306, 13290, 13271,
    13199, 13577, 13185, 13174, 13258, 13153, 13392, 13266, 13022, 13096, 12898, 13160,
    13177, 13244, 12622, 12964, 13011, 12995, 13161, 12716, 12891, 12805, 12817, 13046,
    13093, 12673, 12827, 12725, 12517, 12613, 12658, 12720, 12517, 12926, 12604, 12597,
    12628, 12393, 12757, 12745, 12543, 12775, 12448, 12314, 12284, 12441, 12114, 12493,
    12463, 12195, 12129, 12111, 11949, 12306, 12118, 12351, 12332, 12168, 12141, 12169,
    12000, 11986, 12013, 12142, 12110, 12011, 12265, 11905, 11907, 11792, 12037, 11774,
    11771, 11874, 11840, 12046, 11773, 11636, 11751, 11652, 11786, 11521, 11574, 11619,
    11598, 12056, 11546, 11554, 11867, 11332, 11384, 11535, 11548, 11398, 11517, 11424,
    11398, 11385, 11609, 11297, 11588, 11222, 11452, 11390, 11072, 11121, 11215, 11122,
    10992, 10948, 11319, 11001, 11223, 11348, 10749, 11281, 11036, 10987, 11185, 10986,
    10921, 11003, 10942, 11047, 10876, 10757, 11116, 10654, 10921, 10784, 10846, 10680,
    10653, 10859, 10535, 6965652], dtype=np.int64)
```

```python
wikipedia_1024_max_sequence_length = 1024

wikipedia_2048_histogram = np.array([
       0, 0, 0, 0, 2477, 1876, 3242, 2262, 7312, 2795, 4079, 5706,
    6488, 10440, 11572, 18367, 19043, 17166, 18433, 20247, 22804, 24700, 27419, 30059,
    32627, 35840, 39700, 42465, 45913, 48281, 50135, 53069, 55707, 57654, 60733, 63289,
    65678, 67824, 70064, 72022, 74546, 75868, 77463, 78728, 80340, 80598, 81369, 82172,
    82161, 83038, 82645, 82620, 81833, 81836, 80906, 81093, 81594, 80329, 81265, 81015,
    79730, 79043, 78811, 80007, 78575, 78209, 78174, 77714, 76950, 76864, 75966, 76074,
    74945, 75533, 74347, 73401, 72540, 72503, 71834, 70761, 70221, 68597, 68371, 67307,
    66927, 66421, 65566, 64768, 64117, 63245, 62774, 62196, 61666, 61419, 60865, 59983,
    59731, 58935, 58353, 58432, 57617, 57372, 57232, 56518, 55999, 55816, 55627, 55505,
    54940, 54207, 53537, 53462, 53342, 52812, 52522, 52094, 51834, 51047, 50868, 50703,
    50178, 50507, 50081, 50183, 48968, 49051, 48651, 48129, 47735, 47660, 47069, 47101,
    46740, 46577, 46858, 46588, 46340, 45488, 45065, 45149, 45238, 44779, 45004, 44332,
    43872, 43926, 43603, 43376, 42703, 43093, 42671, 42189, 42130, 41791, 41566, 41341,
    41309, 41411, 40457, 41006, 40225, 40108, 39568, 40082, 39498, 39557, 39608, 39236,
    38730, 38549, 39364, 38165, 38267, 38112, 37755, 37777, 37449, 37474, 37799, 36787,
    36650, 36437, 37130, 36613, 36214, 36071, 36418, 36246, 35613, 35805, 35826, 35031,
    34758, 34993, 34890, 34458, 34690, 34282, 33928, 34027, 34037, 34079, 33932, 33961,
    33894, 33497, 33642, 33634, 33393, 33305, 32561, 33038, 32708, 32127, 32435, 32092,
    32203, 32239, 31599, 32348, 31303, 31696, 31438, 31155, 30889, 30825, 31209, 30380,
    30619, 30494, 30875, 29938, 30435, 29785, 30119, 29787, 29785, 29481, 29369, 29160,
    29134, 29033, 29317, 29069, 28934, 28961, 28603, 28319, 28568, 28798, 28318, 28095,
    28397, 28244, 27782, 27889, 27584, 27322, 27299, 27665, 27066, 26982, 27232, 26753,
    26673, 27066, 26812, 26270, 26036, 26053, 26415, 26086, 25782, 25645, 25719, 25757,
    25630, 25920, 25268, 25639, 25350, 25564, 25032, 25018, 25226, 25065, 24904, 24619,
    24696, 24732, 24269, 24633, 24565, 24257, 24304, 24427, 24043, 23844, 23872, 23869,
    23439, 23613, 23434, 23735, 23325, 23362, 23119, 23373, 23561, 23088, 23213, 23074,
    22859, 22651, 22644, 22570, 22813, 22739, 22704, 22380, 22568, 21998, 22210, 21782,
    22120, 22003, 22079, 22104, 21610, 21464, 21687, 21587, 21167, 21427, 21670, 21336,
    21382, 21465, 21291, 20896, 21016, 20776, 21016, 20613, 20666, 20795, 20830, 20680,
    20213, 20221, 19983, 20175, 20136, 20361, 19928, 19803, 20031, 19887, 19899, 20007,
    19746, 19429, 19800, 19353, 19597, 19708, 19247, 19181, 19396, 19301, 19071, 19292,
    19370, 18672, 18626, 19062, 18839, 19238, 18705, 18741, 18611, 18673, 18649, 18607,
    18288, 18492, 18250, 18295, 18043, 18118, 18065, 18015, 18046, 17872, 18000, 17777,
    17812, 17899, 17832, 17604, 17389, 17259, 17594, 17654, 17632, 17437, 17571, 17444,
    17221, 17363, 17137, 17013, 17228, 16846, 16678, 16901, 17003, 17015, 16700, 16471,
    16574, 16531, 16556, 16363, 16267, 16498, 16513, 16469, 16352, 16434, 16283, 16636,
    16059, 16047, 16299, 15739, 16200, 15832, 16017, 15751, 15870, 15851, 15796, 15845,
    15618, 15675, 15504, 15608, 15358, 15712, 15423, 15366, 15539, 15175, 15122, 15092,
    15435, 15376, 15097, 15012, 14764, 15224, 14700, 14831, 14973, 14906, 14667, 14639,
    14901, 14918, 14416, 14724, 14525, 14643, 14837, 14175, 14598, 14481, 14416, 14192,
    14185, 14256, 14249, 14096, 14393, 14043, 14080, 14034, 14113, 14249, 14066, 14003,
    14089, 13892, 13609, 13920, 13896, 13642, 13703, 13896, 13711, 13631, 13807, 13704,
    13447, 13687, 13535, 13467, 13657, 13624, 13735, 13463, 13257, 13162, 13490, 13377,
    13194, 12986, 13308, 13407, 13192, 12968, 13076, 12980, 13011, 12946, 12851, 12931,
    12768, 12772, 12885, 12939, 12707, 12787, 12675, 12616, 12525, 12386, 12486, 12479,
    12776, 12431, 12297, 12294, 12252, 12404, 12387, 12421, 12540, 12010, 12297, 12285,
    12252, 12021, 12042, 11944, 12016, 11910, 11914, 11931, 12013, 11687, 11610, 11493,
    12047, 11580, 11890, 11661, 11707, 11683, 11551, 11449, 11450, 11127, 11488, 11366,
    11109, 11150, 11363, 11258, 11165, 11156, 11097, 11304, 11144, 11264, 11243, 11068,
    11027, 11066, 11078, 11035, 10973, 10845, 11028, 10871, 10822, 10974, 10817, 10619,
    10532, 10617, 10635, 10513, 10625, 10725, 10434, 10293, 10630, 10616, 10607, 10293,
    10603, 10244, 10304, 10439, 10228, 10325, 10331, 9887, 9972, 10385, 10159, 10089,
    10112, 10180, 10213, 10078, 10138, 9937, 9914, 10042, 9899, 9845, 9716, 10107,
    9889, 9861, 9703, 9578, 9722, 9757, 9713, 9483, 9572, 9676, 9911, 9636,
    9429, 9723, 9657, 9613, 9581, 9546, 9432, 9247, 9398, 9384, 9392, 9558,
    9428, 9302, 9269, 9287, 9215, 9296, 9316, 9361, 9265, 9159, 9117, 9127,
    8953, 8952, 9313, 9017, 9087, 8864, 9129, 8895, 9127, 8863, 8791, 8972,
    8686, 8998, 9047, 8895, 8797, 8832, 8752, 8644, 8644, 8755, 8766, 8752,
    8529, 8637, 8476, 8515, 8595, 8407, 8506, 8600, 8572, 8566, 8521, 8514,
    8430, 8272, 8322, 8147, 8112, 8172, 8208, 8233, 8403, 8145, 8153, 8327,
    8233, 8226, 8158, 8207, 8155, 8290, 8200, 8215, 7933, 7882, 8198, 8086,
    7958, 7994, 8204, 8064, 8010, 7944, 7959, 7854, 7768, 7788, 7863, 7766,
    7983, 7895, 7801, 7896, 7811, 7794, 7718, 7670, 7657, 7702, 7602, 7694,
    7877, 7581, 7640, 7599, 7691, 7570, 7484, 7719, 7326, 7551, 7495, 7555,
    7447, 7367, 7345, 7423, 7359, 7357, 7690, 7451, 7369, 7310, 7372, 7301,
    7219, 7374, 7242, 7140, 7381, 7216, 7179, 7042, 7172, 7122, 7170, 7176,
    7165, 7284, 7140, 7074, 7026, 7141, 7016, 7087, 7069, 6851, 6961, 6866,
    6788, 6892, 6990, 6810, 6911, 6850, 6917, 7124, 7012, 6825, 6878, 6719,
    6860, 6842, 6785, 6895, 6929, 6935, 6679, 6625, 6672, 6682, 6818, 6517,
    6768, 6704, 6690, 6651, 6477, 6465, 6530, 6708, 6521, 6634, 6597, 6622,
    6594, 6361, 6337, 6509, 6548, 6393, 6515, 6188, 6347, 6321, 6408, 6407,
    6230, 6310, 6112, 6294, 6297, 6110, 6284, 6340, 6202, 6147, 6213, 6236,
    6259, 6260, 6160, 6276, 6002, 6096, 6166, 6239, 5964, 6007, 6042, 6173,
    6242, 6279, 6004, 6297, 6035, 6039, 5945, 5859, 6062, 6017, 5894, 6016,
    5958, 6012, 6110, 5839, 5836, 5794, 5858, 5947, 5753, 5829, 5633, 5920,
    5834, 5885, 5649, 5744, 5696, 5854, 5698, 5761, 5742, 5972, 5736, 5747,
    5777, 5720, 5739, 5648, 5620, 5565, 5459, 5592, 5655, 5577, 5674, 5562,
    5696, 5645, 5566, 5626, 5342, 5838, 5606, 5461, 5474, 5484, 5332, 5429,
    5560, 5476, 5466, 5262, 5270, 5457, 5389, 5459, 5449, 5307, 5334, 5289,
    5324, 5335, 5314, 5222, 5223, 5462, 5392, 5255, 5306, 5139, 5196, 5194,
    5367, 5287, 5224, 5218, 5229, 5234, 5107, 5241, 5077, 5049, 5173, 5157,
    5084, 5070, 5171, 5057, 5065, 5046, 4988, 5045, 5016, 4988, 5043, 5086,
    4982, 5013, 4932, 4938, 4965, 4942, 5004, 4887, 4896, 4783, 4991, 4984,
    4875, 4805, 4995, 4865, 4866, 4890, 4627, 4921, 4745, 4734, 4781, 4970,
    4696, 4759, 4639, 4791, 4805, 4896, 4852, 4671, 4937, 4739, 4584, 4671,
    4662, 4678, 4770, 4702, 4605, 4751, 4626, 4604, 4603, 4631, 4798, 4599,
    4658, 4744, 4571, 4493, 4609, 4480, 4632, 4641, 4625, 4440, 4512, 4491,
    4401, 4562, 4661, 4542, 4597, 4663, 4494, 4553, 4553, 4504, 4349, 4425,
    4456, 4366, 4405, 4300, 4329, 4501, 4508, 4415, 4333, 4348, 4290, 4360,
```

```
            4356, 4202, 4337, 4254, 4262, 4323, 4176, 4374, 4436, 4300, 4415, 4316,
            4342, 4316, 4329, 4189, 4177, 4206, 4387, 4266, 4103, 4227, 4227, 4214,
            4238, 4126, 4193, 4159, 4089, 4115, 4215, 4087, 4099, 4064, 4139, 4085,
            4160, 4074, 4130, 4031, 4099, 4143, 4129, 4021, 4152, 4048, 4025, 4117,
            3966, 3833, 4059, 4044, 4081, 4051, 3990, 3979, 3987, 3924, 4025, 3934,
            3961, 3911, 3993, 3927, 4055, 3865, 3935, 4005, 3894, 3852, 3997, 3990,
            3869, 3898, 3853, 3866, 3888, 3992, 3764, 3812, 3886, 3676, 3794, 3904,
            3957, 3852, 3848, 3746, 3832, 3834, 3751, 3797, 3750, 3656, 3853, 3776,
            3764, 3680, 3632, 3695, 3635, 3715, 3677, 3610, 3818, 3619, 3675, 3652,
            3806, 3787, 3738, 3620, 3677, 3575, 3736, 3679, 3724, 3754, 3609, 3613,
            3643, 3701, 3558, 3698, 3660, 3651, 3586, 3437, 3513, 3623, 3551, 3580,
            3532, 3506, 3528, 3614, 3508, 3483, 3405, 3514, 3590, 3451, 3516, 3405,
            3417, 3554, 3454, 3595, 3410, 3411, 3496, 3550, 3586, 3498, 3518, 3438,
            3407, 3446, 3589, 3343, 3420, 3195, 3455, 3329, 3368, 3356, 3502, 3482,
            3349, 3456, 3348, 3388, 3362, 3371, 3316, 3251, 3349, 3441, 3419, 3311,
            3430, 3306, 3359, 3236, 3151, 3232, 3285, 3295, 3252, 3126, 3236, 3323,
            3331, 3203, 3190, 3180, 3303, 3203, 3137, 3155, 3256, 3206, 3155, 3096,
            3162, 3160, 3223, 3140, 3262, 3176, 3189, 3247, 3208, 3242, 3217, 3131,
            3113, 3235, 3119, 3196, 3130, 3052, 3150, 3093, 3234, 3115, 3059, 3376,
            3171, 3195, 3082, 3051, 3106, 3026, 2983, 3125, 3062, 3049, 3205, 3001,
            2948, 3110, 2881, 2987, 2950, 3091, 2994, 2965, 3099, 3069, 2984, 2977,
            2967, 2988, 2928, 3071, 2986, 2999, 2937, 3089, 2883, 2991, 2927, 3060,
            2806, 3004, 2856, 2876, 2935, 2944, 2864, 2880, 2903, 2782, 2747, 2916,
            3015, 2928, 3012, 2857, 2909, 2806, 2863, 2883, 2806, 2878, 2928, 2803,
            2850, 2846, 2746, 2814, 2865, 2815, 2788, 2906, 2810, 2789, 2787, 2705,
            2825, 2803, 2926, 2807, 2765, 2797, 2747, 2796, 2683, 2780, 2844, 2848,
            2809, 2825, 2611, 2739, 2717, 2642, 2664, 2757, 2807, 2704, 2809, 2689,
            2684, 2828, 2637, 2722, 2647, 2745, 2714, 2717, 2784, 2732, 2570, 2687,
            2677, 2653, 2796, 2619, 2647, 2568, 2727, 2642, 2672, 2603, 2578, 2807,
            2815, 2665, 2623, 2661, 2605, 2685, 2562, 2573, 2616, 2594, 2625, 2515,
            2658, 2464, 2624, 2564, 2637, 2698, 2572, 2631, 2527, 2622, 2586, 2535,
            2502, 2574, 2554, 2584, 2565, 2542, 2547, 2520, 2398, 2593, 2699, 2474,
            2355, 2496, 2492, 2533, 2558, 2582, 2424, 2465, 2540, 2470, 2531, 2566,
            2391, 2540, 2556, 2405, 2519, 2495, 2557, 2544, 2561, 2414, 2528, 2536,
            2521, 2468, 2458, 2408, 2524, 2397, 2477, 2286, 2278, 2503, 2469, 2385,
            2400, 2435, 2376, 2416, 2346, 2425, 2393, 2364, 2373, 2314, 2359, 2384,
            2397, 2409, 2372, 2491, 2296, 2412, 2236, 2413, 2420, 2400, 2379, 2471,
            2403, 2421, 2270, 2389, 2290, 2371, 2284, 2363, 2381, 2409, 2245, 2228,
            2391, 2304, 2248, 2270, 2367, 2282, 2236, 2361, 2168, 2305, 2353, 2260,
            2244, 2323, 2255, 2409, 2219, 2293, 2324, 2262, 2303, 2301, 2195, 2302,
            2293, 2188, 2189, 2255, 2173, 2254, 2094, 2225, 2165, 2276, 2283, 2317,
            2217, 2136, 2299, 2270, 2288, 2112, 2266, 2118, 2270, 2204, 2110, 2278,
            2215, 2227, 2131, 2215, 2255, 2238, 2129, 2141, 2203, 2054, 2171, 2170,
            2132, 2162, 2069, 2290, 2221, 2122, 2208, 2121, 2134, 2120, 2137, 2172,
            2165, 2195, 2100, 2044, 1985, 2058, 2104, 2037, 2126, 2121, 2043, 1994,
            2102, 2114, 2003, 2069, 2055, 2120, 2080, 2098, 2058, 2021, 2049, 2097,
            2162, 2195, 2022, 2146, 2084, 2047, 2006, 2009, 2181, 2039, 2059, 2053,
            1987, 1995, 2105, 2006, 1967, 2046, 2005, 2049, 2050, 2139, 2068, 1968,
            1929, 2058, 1997, 2050, 2092, 1922, 1976, 2023, 2065, 2003, 1976, 2027,
            1978, 2052, 1978, 2005, 1997, 1972, 1990, 2033, 2035, 1931, 2012, 2009,
            1890, 1900, 1879, 1946, 2078, 1976, 2011, 1916, 1963, 2058, 1998, 1906,
            1964, 1937, 1884, 1970, 1967, 1913, 1853, 1843, 1985, 1912, 1931, 1932,
            1903, 1878, 1915, 1886, 1941, 1899, 1840, 1767, 1889, 1862, 1986, 1923,
            1908, 1868, 1913, 1797, 1773, 1871, 1780, 1815, 1951, 1840, 1787, 1920,
            1909, 1835, 1932, 1826, 1944, 1819, 1831, 1865, 1818, 1829, 1837, 1889,
            1809, 1834, 1845, 1824, 1911, 1910, 1842, 1760, 1837, 1875, 1838, 1804,
            1713, 1801, 1779, 1713, 1864, 1899, 1802, 1799, 1859, 1772, 1884, 1797,
            1827, 1751, 1738, 1683, 1725, 1816, 1733, 1775, 1761, 1771, 1824, 1860,
            1775, 1827, 1808, 1760, 1691, 1694, 1753, 1759, 1744, 1750, 1742, 1801,
            1783, 1832, 1737, 1764, 1755, 1788, 1764, 1730, 1777, 1761, 1724, 1707,
            1796, 1726, 1739, 1717, 1754, 1789, 1719, 1694, 1651, 1762, 1693, 1717,
            1717, 1750, 1697, 1685, 1681, 1684, 1709, 1745, 1707, 1641, 1649, 1710,
            1638, 1670, 1728, 1662, 1625, 1731, 1657, 1620, 1746, 1746, 1726, 1659,
            1686, 1637, 1653, 1615, 1650, 1712, 1616, 1621, 1581, 1649, 1646, 1687,
            1748, 1749, 1614, 1629, 1636, 1729, 1568, 1661, 1638, 1614, 1545, 1660,
            1642, 1677, 1614, 1627, 1572, 1675, 1725, 1694, 1638, 1613, 1570, 1540,
            1644, 1527, 1622, 1584, 1646, 1512, 1619, 1534, 1644, 1613, 1584, 1488,
            1612, 1469, 1624, 1550, 1487, 1524, 1569, 1570, 1563, 1552, 1572, 1526,
            1574, 1511, 1673, 1557, 1521, 1495, 1502, 1658, 1547, 1602, 1541, 1617,
            1440, 1545, 1528, 1610, 1483, 1583, 1511, 1601, 1564, 1527, 1501, 1451,
            1588, 1485, 1555, 1541, 1468, 1430, 1464, 1517, 1569, 1541, 1521, 1538,
            1417, 1502, 1491, 1522, 1518, 1486, 1537, 1413, 1572, 1492, 1456, 1396,
            1517, 1471, 1422, 1494, 1406, 1510, 1512, 1495, 1536, 1454, 1429, 1494,
            1485, 1489, 1525, 1529, 1562, 1461, 1500, 1450, 1409, 1428, 1509, 1509,
            1509, 1414, 1471, 1500, 1361, 1481, 1444, 1470, 1520, 1458, 1463, 1465,
            1484, 1439, 1386, 1463, 1379, 1482, 1396, 1441, 1405, 1495, 1551, 1473,
            1389, 1385, 1360, 1417, 1379, 1435, 1445, 1372, 1483, 1349, 1441, 1353,
            1538, 1370, 1401, 1421, 1414, 1493, 1418, 1363, 1372, 1303, 1397, 1411,
            1325, 1436, 1382, 1421, 1384, 1391, 1471, 1472, 1431, 1440, 1413, 1399,
            1361, 1375, 1341, 1379, 1420, 1402, 1338, 1334, 1405, 1390, 1370, 1463,
            1344, 1456, 1444, 1379, 1401, 1372, 1334, 1406, 1355, 1343, 1377, 1376,
            1382, 1341, 1337, 1385, 1322, 1380, 1286, 1503796], dtype=np.int64)

wikipedia_2048_max_sequence_length = 2048

squad_1_1_histogram = np.array([
        0, 0, 0, 0, 0, 0, 0, 0, 0, 0, 0, 0, 0, 0, 0, 0, 0, 0, 0, 0, 0, 0, 0, 0, 0, 0, 0, 0, 0, 0, 0, 0, 0,
        0, 0, 3, 2, 0, 9, 10, 16, 22, 24, 36, 35, 46, 42, 48, 57, 86, 83, 86, 87, 86, 97, 90, 99, 85, 94,
        105, 114, 110, 93, 116, 118, 114, 116, 117, 127, 115, 155, 137, 145, 157, 151, 153, 149, 163, 157,
        134, 150, 144, 132, 166, 162, 177, 160, 149, 151, 138, 156, 148, 176, 163, 182, 188, 182, 177, 199,
        182, 203, 201, 264, 250, 244, 289, 346, 327, 298, 377, 386, 444, 431, 503, 553, 532, 570, 611, 677,
        648, 673, 712, 722, 745, 692, 697, 747, 754, 741, 777, 781, 825, 813, 836, 777, 776, 756, 789, 790,
```

```
   765, 753, 729, 748, 772, 766, 760, 741, 725, 729, 759, 732, 730, 730, 741, 705, 708, 725, 656, 688,
   688, 677, 662, 628, 635, 618, 586, 527, 562, 619, 562, 578, 538, 558, 582, 541, 575, 526, 556, 498,
   529, 486, 528, 541, 482, 521, 483, 466, 514, 459, 447, 436, 383, 401, 408, 381, 369, 364, 381, 420,
   391, 388, 358, 365, 357, 358, 355, 297, 290, 267, 308, 329, 304, 332, 289, 282, 304, 242, 263, 288,
   238, 257, 271, 288, 277, 264, 253, 239, 217, 260, 214, 247, 237, 212, 205, 193, 200, 208, 195, 193,
   201, 187, 170, 176, 195, 156, 201, 179, 159, 183, 169, 178, 163, 153, 171, 144, 138, 181, 165, 171,
   161, 159, 166, 142, 138, 151, 155, 134, 141, 132, 123, 119, 109, 125, 123, 131, 135, 115, 108, 102,
   117, 105, 99, 84, 100, 85, 85, 85, 95, 122, 105, 114, 113, 100, 80, 96, 86, 79, 80, 87, 92, 73, 73,
   64, 76, 72, 77, 67, 60, 71, 77, 79, 72, 55, 67, 42, 59, 65, 72, 49, 43, 62, 48, 50, 54, 45, 42, 53,
   56, 45, 43, 32, 30, 36, 42, 37, 45, 28, 41, 31, 44, 35, 36, 47, 47, 48, 65, 32, 23, 35, 38, 20, 23,
   22, 21, 27, 20, 26, 18, 18, 22, 17, 17, 14, 26, 15, 20, 22, 19, 24, 17, 15, 20, 20, 22, 22, 17, 20,
   16, 21, 16, 23, 12, 14, 1054], dtype=np.int64)

squad_1_1_max_sequence_length = 384
```

Listing 7: Histogram creation for GLUE training datasets

```python
# Copyright 2020 The HuggingFace Inc. team. All rights reserved.
#
# Licensed under the Apache License, Version 2.0 (the "License");
# you may not use this file except in compliance with the License.
# You may obtain a copy of the License at
#
#     http://www.apache.org/licenses/LICENSE-2.0
#
# Unless required by applicable law or agreed to in writing, software
# distributed under the License is distributed on an "AS IS" BASIS,
# WITHOUT WARRANTIES OR CONDITIONS OF ANY KIND, either express or implied.
# See the License for the specific language governing permissions and
# limitations under the License.
"""GLUE data loading and histogram creation.

Some code snippets were taken from
https://github.com/huggingface/transformers/blob/master/examples/pytorch/text-classification/run_glue.py
Most is original code.
"""
from transformers import AutoTokenizer
import datasets
import numpy as np

# constants
max_sequence_length = 128
task_to_keys = {
    "cola": ("sentence", None),
    "mnli": ("premise", "hypothesis"),
    "mrpc": ("sentence1", "sentence2"),
    "qnli": ("question", "sentence"),
    "qqp": ("question1", "question2"),
    "rte": ("sentence1", "sentence2"),
    "sst2": ("sentence", None),
    "stsb": ("sentence1", "sentence2"),
    "wnli": ("sentence1", "sentence2"),
}
glue_keys = ['cola', 'sst2', 'mrpc', 'qqp', 'stsb', 'mnli', 'rte', 'wnli']
# unused datasets due to missing training data
unglue_keys = ['mnli_matched', 'mnli_mismatched', 'qnli', 'ax']

# load data
dataset_loads = {}
for key in glue_keys:
    dataset_loads[key] = datasets.load_dataset("glue", key, split='train')

# tokenize data
tokenizer = AutoTokenizer.from_pretrained('bert-base-uncased')
tokenized_data = {}
for key in dataset_loads:
    sentence1_key, sentence2_key = task_to_keys[key]

    def preprocess_function(examples):
        """Tokenize the texts"""
        args = (
            (examples[sentence1_key],) if sentence2_key is None
            else (examples[sentence1_key], examples[sentence2_key])
        )
        result = tokenizer(*args, padding=False, max_length=max_sequence_length, truncation=True)
        return result

    tokenized_data[key] = dataset_loads[key].map(preprocess_function, batched=True)

# extract length information (for histogram plots)
histogram_length = {}
for key in tokenized_data:
    histogram_length[key] = []
for number, key in enumerate(tokenized_data.keys()):
    for raw_record in tokenized_data[key]["input_ids"]:
        histogram_length[key].append(len([x for x in raw_record if x!=0]))

# create histogram for packing
glue_histogram = {}
for data_key in histogram_length:
    glue_histogram[data_key] = np.array([0] * max_sequence_length, dtype=np.int64)
    for entry in histogram_length[data_key]:
        glue_histogram[data_key][entry-1] += 1
```

Listing 8: Longest-pack-first histogram-packing

```python
from collections import defaultdict
import numpy as np
import time

def add_pack(pack, count, tmp, final, limit, offset, max_sequence_length=512):
    """Filter out packs that reached maximum length or number of components."""
    # sanity checks
    assert(max_sequence_length-sum(pack) == offset), "Incorrect offset."
    assert(offset >= 0), "Too small offset."
    assert(offset < max_sequence_length), "Too large offset."
    if len(pack) == limit or offset == 0:
        final[offset].append((count, pack))
    else:
        tmp[offset].append((count, pack))

def pack_using_lpfhp(histogram, max_sequence_length, max_sequences_per_pack, distribute=True):
    """Longest-pack-first histogram-packing."""
    start = time.time()
    reversed_histogram = np.flip(histogram)
    # Initialize main strategy data dictionary.
    # The key indicates how many tokens are left for full length.
    # The value is a list of tuples, consisting of counts and respective packs.
    # A pack is a (sorted) list of sequence length values that get concatenated.
    tmp_strategies_per_length = defaultdict(list)
    strategies_per_length = defaultdict(list)
    if max_sequences_per_pack is "max":
        max_sequences_per_pack = max_sequence_length
    # Index i indicates here, how much space is left, due to reversed histogram
    for i in range(max_sequence_length):
        n_sequences_to_bin = reversed_histogram[i]
        length_to_bin = max_sequence_length - i
        offset = 0  # smallest possible offset for perfect fit
        while n_sequences_to_bin > 0:
            if (length_to_bin + offset) in tmp_strategies_per_length:
                # extract worst pack that will get modified
                n_sequences_to_pack, pack = tmp_strategies_per_length[
                    length_to_bin + offset].pop()
                # calculate how often the current sequence maximally fits in
                repeat = min(1 + offset // length_to_bin, max_sequences_per_pack-len(pack))
                # correct dependent on count
                while n_sequences_to_bin//repeat == 0:
                    repeat -= 1
                if not distribute:
                    repeat = 1
                new_pack = pack + [length_to_bin]*repeat
                count = min(n_sequences_to_pack, n_sequences_to_bin//repeat)
                if n_sequences_to_pack > count:
                    # old pack gets reduced
                    n_sequences_to_pack -= count
                    tmp_strategies_per_length[length_to_bin + offset].append(
                        (n_sequences_to_pack, pack))
                    n_sequences_to_bin -= count * repeat
                else:
                    n_sequences_to_bin -= n_sequences_to_pack * repeat
                add_pack(new_pack, count,
                         tmp_strategies_per_length, strategies_per_length,
                         max_sequences_per_pack, offset - (repeat - 1) * length_to_bin,
                         max_sequence_length)
                # clean up to speed up main key search
                if not tmp_strategies_per_length[length_to_bin + offset]:
                    tmp_strategies_per_length.pop(length_to_bin + offset)
                # reset offset in case best fit changed
                offset = 0
            else:
                offset += 1
            # Does not fit anywhere. Create new pack.
            if offset >= max_sequence_length - length_to_bin + 1:
                # similar repetition but no dependence on pack.
                repeat = min(max_sequence_length//length_to_bin, max_sequences_per_pack)
                while n_sequences_to_bin//repeat == 0:
                    repeat -= 1
                if not distribute:
                    repeat = 1
                add_pack([length_to_bin]*repeat, n_sequences_to_bin//repeat,
                         tmp_strategies_per_length, strategies_per_length,
                         max_sequences_per_pack, max_sequence_length-length_to_bin*repeat,
                         max_sequence_length)
                n_sequences_to_bin -= n_sequences_to_bin//repeat * repeat
```

```python
    # merge all strategies
    for key in tmp_strategies_per_length:
        strategies_per_length[key].extend(tmp_strategies_per_length[key])
    # flatten strategies dictionary
    strategy_set = []
    strategy_repeat_count = []
    for key in strategies_per_length:
        for count, pack in strategies_per_length[key]:
            pack.reverse()
            strategy_set.append(pack)
            strategy_repeat_count.append(count)

    # Summarize efficiency of solution
    duration = time.time() - start
    sequence_lengths = np.arange(1, max_sequence_length + 1)
    strategy_repeat_count = np.array(strategy_repeat_count)
    n_strategies = len(strategy_set)
    old_number_of_samples = histogram.sum()
    new_number_of_samples = strategy_repeat_count.sum()
    sequences = sum([count*len(pack) for count, pack in
                    zip(strategy_repeat_count, strategy_set)])
    total_tokens = max_sequence_length * new_number_of_samples
    empty_tokens = sum([count*(max_sequence_length-sum(pack)) for count, pack
                    in zip(strategy_repeat_count, strategy_set)])
    efficiency = 100 - empty_tokens / total_tokens * 100
    speedup_upper_bound = 1.0/(1 - (histogram*(
        1 - sequence_lengths / max_sequence_length)).sum() / old_number_of_samples)

    print(f"Packing efficiency (fraction of real tokens): {efficiency:3.4f}\n",
          f"Speed-up theoretical limit: {speedup_upper_bound:3.4f}\n",
          f"Achieved speed-up over un-packed dataset: {old_number_of_samples/new_number_of_samples:3.5f}",
          f"Runtime: Packed {old_number_of_samples} sequences in {duration:3.3f} seconds.")

    return strategy_set, strategy_repeat_count
```

## Listing 9: Extended non-negative least squares histogram-packing

```python
import time
import numpy as np
from scipy import optimize, stats
from functools import lru_cache

def get_packing_matrix(strategy_set, max_sequence_length):
    num_strategies = len(strategy_set)
    A = np.zeros((max_sequence_length, num_strategies), dtype=np.int32)
    for i, strategy in enumerate(strategy_set):
        for seq_len in strategy:
            A[seq_len - 1, i] += 1
    return A

@lru_cache(maxsize=None)
def get_packing_strategies(start_length, minimum_increment, target_length, depth):
    gap = target_length - start_length
    strategies = []
    # Complete the packing with exactly 1 number
    if depth == 1:
        if gap >= minimum_increment:
            strategies.append([gap])
    # Complete the sample in "depth" steps, recursively
    else:
        for new in range(minimum_increment, gap + 1):
            new_gap = target_length - start_length - new
            if new_gap == 0:
                strategies.append([new])
            else:
                options = get_packing_strategies(start_length + new, new, target_length, depth - 1)
                for option in options:
                    if len(option) > 0:
                        strategies.append([new] + option)
    return strategies

def pack_using_ennlshp(histogram, max_sequence_length, max_sequences_per_pack):
    # List all unique ways of packing to the desired maximum sequence length
    strategy_set = get_packing_strategies(0, 1, max_sequence_length, max_sequences_per_pack)
    print(f"Packing will involve {len(strategy_set)} unique packing strategies.")
    # Get the packing matrix corresponding to this list of packing strategies
    A = get_packing_matrix(strategy_set, max_sequence_length)
    # Weights that penalize the residual by the number of resulting padding tokens.
    w0 = np.array([x+1 for x in range(max_sequence_length)])
    # construct the packing matrix
    A_bar = np.zeros((2*max_sequence_length, len(strategy_set) + max_sequence_length), 'd')
    # Base weighted matrix
    A_bar[:max_sequence_length, :len(strategy_set)] = np.expand_dims(w0, -1) * A
    # Higher weight to avoid positive residual
    A_bar[max_sequence_length:, :len(strategy_set)] = np.expand_dims(
        10**6*np.ones([max_sequence_length]), -1) * A
    # negative diagonal unity matrix for mapping to residual
    A_bar[max_sequence_length:, len(strategy_set):] = np.expand_dims(
        10**6*np.ones([max_sequence_length]), -1)*np.ones((max_sequence_length,max_sequence_length))
    b_bar = np.zeros(2*max_sequence_length)
    # Apply weighting to histogram vector
    b_bar[:max_sequence_length] = w0 * histogram
    b_bar[max_sequence_length:] = 10**6*np.ones([max_sequence_length]) * histogram
    # Solve the packing problem
    print(f"Sequences to pack: ", histogram.sum())
    start = time.time()
    strategy_residual, rnorm = optimize.nnls(A_bar, b_bar)
    strategy_repeat_count = strategy_residual[:len(strategy_set)]
    print(f"Solving non-negative least squares took {time.time() - start:3.2f} seconds.")
    # Round the floating point solution to nearest integer
    strategy_repeat_count = np.rint(strategy_repeat_count).astype(np.int64)
    # Compute the residuals, shape: [max_sequence_length]
    residual = histogram - A @ strategy_repeat_count
    # Handle the left-over sequences i.e. positive part of residual
    unpacked_seqlen = np.arange(1, max_sequence_length + 1)[residual > 0]
    for l in unpacked_seqlen:
        strategy = sorted([l, max_sequence_length - l])  # the depth 1 strategy
        strategy_index = strategy_set.index(strategy)
        strategy_repeat_count[strategy_index] += residual[l-1]
    # Re-compute the residual with the updated strategy_repeat_count
    # This should now be strictly < 0
    residual = histogram - A @ strategy_repeat_count
    # Add padding based on deficit (negative residual portion of residual)
    padding = np.where(residual < 0, -residual, 0)
    # Calculate some basic statistics
    sequence_lengths = np.arange(1, max_sequence_length + 1)
    old_number_of_samples = histogram.sum()
    new_number_of_samples = int(strategy_repeat_count.sum())
    speedup_upper_bound = 1.0/(1 - (histogram*(
        1 - sequence_lengths / max_sequence_length)).sum()/old_number_of_samples)
    num_padding_tokens_packed = (sequence_lengths * padding).sum()
    efficiency = 1 - num_padding_tokens_packed/(new_number_of_samples*max_sequence_length)
    print(f"Packing efficiency (fraction of real tokens): {efficiency:3.4f}\n",
          f"Speed-up theoretical limit: {speedup_upper_bound:3.4f}\n",
          f"Achieved speed-up over un-packed dataset: {old_number_of_samples/new_number_of_samples:3.5f}")
    return strategy_set, strategy_repeat_count
```

## Appendix References

[37]  BELOV, G., AND SCHEITHAUER, G.   A branch-and-cut-and-price algorithm for one-dimensional stock cutting and two-dimensional two-stage cutting. *European Journal of Operational Research 171*, 1 (may 2006), 85–106.

[38]  GAREY, M. R., AND JOHNSON, D. S. *Computers and Intractability; A Guide to the Theory of NP-Completeness*. W. H. Freeman & Co., USA, 1990.

[39]  GUILLÉN, G., DIAZ-CAMINO, C., LOYOLA-TORRES, C., APARICIO-FABRE, R., HERNÁNDEZ-LÓPEZ, A., DÍAZ-SÁNCHEZ, M., AND SANCHEZ, F. Detailed analysis of putative genes encoding small proteins in legume genomes. *Frontiers in Plant Science 4* (2013), 208.

[40]  HANSEN, H. B., DAMGAARD, P. B., MARGARYAN, A., STENDERUP, J., LYNNERUP, N., WILLERSLEV, E., AND ALLENTOFT, M. E. Comparing ancient dna preservation in petrous bone and tooth cementum. *PLOS ONE 12*, 1 (01 2017), 1–18.

[41]  KOTZ, S., AND NADARAJAH, S. *Extreme Value Distributions*. World Scientific Publishing Company, 2000.

[42]  LAWSON, C. L., AND HANSON, R. J. *Solving Least Squares Problems*. Society for Industrial and Applied Mathematics, jan 1995.

[43]  LUO, Y., AND DURAISWAMI, R. Efficient parallel non-negative least squares on multi-core architectures. *SIAM Journal on Scientific Computing 33* (2011), 2848 – 2863.

[44]  NVIDIA.   Performance catalogue for BERT on Pytorch.   https://ngc.nvidia.com/catalog/resources/nvidia:bert_for_pytorch/performance, 2021.

[45]  PENG, Y., YAN, S., AND LU, Z. Transfer Learning in Biomedical Natural Language Processing: An Evaluation of BERT and ELMo on Ten Benchmarking Datasets. In *Proceedings of the 2019 Workshop on Biomedical Natural Language Processing (BioNLP 2019)* (2019), pp. 58–65.

[46]  VIRTANEN, P., GOMMERS, R., OLIPHANT, T. E., HABERLAND, M., REDDY, T., COURNAPEAU, D., BUROVSKI, E., PETERSON, P., WECKESSER, W., BRIGHT, J., VAN DER WALT, S. J., BRETT, M., WILSON, J., MILLMAN, K. J., MAYOROV, N., NELSON, A. R. J., JONES, E., KERN, R., LARSON, E., CAREY, C. J., POLAT, İ., FENG, Y., MOORE, E. W., VANDERPLAS, J., LAXALDE, D., PERKTOLD, J., CIMRMAN, R., HENRIKSEN, I., QUINTERO, E. A., HARRIS, C. R., ARCHIBALD, A. M., RIBEIRO, A. H., PEDREGOSA, F., VAN MULBREGT, P., AND SCIPY 1.0 CONTRIBUTORS. SciPy 1.0: Fundamental Algorithms for Scientific Computing in Python. *Nature Methods 17* (2020), 261–272.

[47]  WOLF, T., LHOEST, Q., VON PLATEN, P., JERNITE, Y., DRAME, M., PLU, J., CHAUMOND, J., DELANGUE, C., MA, C., THAKUR, A., PATIL, S., DAVISON, J., SCAO, T. L., SANH, V., XU, C., PATRY, N., MCMILLAN-MAJOR, A., BRANDEIS, S., GUGGER, S., LAGUNAS, F., DEBUT, L., FUNTOWICZ, M., MOI, A., RUSH, S., SCHMIDD, P., CISTAC, P., MUŠTAR, V., BOUDIER, J., AND TORDJMANN, A. Datasets. *GitHub. Note: https://github.com/huggingface/datasets 1* (2020).

[48]  WOLFRAM RESEARCH INC. Mathematica, Version 12.2. Champaign, IL, 2020.