# OpenReview forum: "Efficient Sequence Packing without Cross-contamination: Accelerating Large Language Models without Impacting Performance"
_NeurIPS.cc/2022/Conference — NeurIPS 2022 Submitted_

### Official Review · Reviewer_ysRz · 2022-07-04

**Rating:** 8
**Confidence:** 3
**Soundness:** 4 excellent
**Presentation:** 4 excellent
**Contribution:** 4 excellent

**Summary:**

This paper describes an approach for packing sequences into batches in order to improve the training efficiency of large-scale language models (and related tasks).
First, several datasets are analyzed with respect to their sequence length variations and the potential speed-up if batches were packed with minimized padding. The paper then formalizes the packing problem and proposes two new algorithms, that in turn require slight model adaptations for the example of BERT. In experiments on BERT pre-training, they showcase the packing efficiency on IPUs and resulting realized speed-up in comparison to other packing (or none) algorithms, and investigate required hyperparameter changes on the MLPerf benchmark. An ablation study of the model adaptations of BERT confirm that these are indeed necessary, and that packed pretraining does not negatively affect fine-tuning (non-packed) on the SQuAD benchmark.

**Questions:**

1. Which BERT pre-processing is referred to in Section 1 (line 23)?

**Limitations:**

The authors mention more challenging applications in computer vision as a next step, and potential difficulties with the implementation. I'd appreciate if the limitations of the current approach would be discussed in more depth: are there any edge cases where the overhead would be prohibitively large? In which data conditions should this packing approach be avoided?

**Strengths And Weaknesses:**

Strengths:
1. The paper sheds light on a problem that has previously been hidden in NLP implementations and has not been clearly documented/researched/discussed, despite its effect on required training resources, hyperparameters, and model quality. The paper is educational for anyone who has not delved deeply into this topic or has not implemented packing themselves (like myself included), and brings transparency into the choice of packing implementations and their impact.
2. As the paper outlines in Section 2, the potential gain of efficiency is large, since sequence length varies a lot in many NLP tasks. The proposed solution is therefore a very impactful contribution to NLP implementations, thereby the efficiency of NLP research.
3. The empirical results are strong, with theoretical speed-ups being close to be realized with little overhead.

Weaknesses:
1. Packing is only applied during pre-training, not during fine-tuning. Adding an experiment on top of the existing SQuAD experiments where packing is continued during fine-tuning would make the proposed approach more convincing, especially since Section 2 lists fine-tuning datasets for motivation.
2. The experimental conditions for 4.1 are not well introduced. Is it BERT pre-training?

---

> ### Author Response · Authors · 2022-08-01
> **Response to Reviewer ysRz**
>
> We thank the reviewer for being as excited about the paper as we are. The comments about the educational value, the significant impact on a variety of applications, and the strong empirical results are reassuring and an important reason to give us the opportunity to present this paper at a major conference.
>
> Let us firstly **clarify the weaknesses** addressed by the reviewer:
>
> - SQuAD fine-tuning: Pre-training BERT on Wikipedia takes much longer than fine-tuning it on SQuAD (more than 100 times). Hence, the benefits of packing in fine-tuning are smaller than 0.5% of the overall speed-up. Our main objective for using the dataset was not to address fine-tuning, but to show that speed-ups can be expected over a variety of applications/datasets. There are 3 potential workarounds to address this in the paper and we would like to hear, which one is the preferred one for the reviewer. For now, we remove the reference to fine-tuning. An alternative is to remove the dataset from Section 2, since we are already analysing a large amount of speech related datasets. If despite the minimal impact, the evaluation of the acceleration of packing on fine-tuning is of interest, we could implement the approach. SQuAD has different loss functions, which require some coding effort. Whereas we follow the concepts described in the paper, we spent some engineering effort to keep the impact of packing on the computational workload very minimal. Thus, a new implementation for each fine-tuning task would be required.
>
> - Experiment description for Section 4.1: Thanks for catching this error caused by numerous content rearrangements. The setup is BERT pre-training and only phase 2 with sequence length 512. We address it by adding additional details to the experiment description and also reference section 4.2 which has the same setup.
>
> Let us now **answer the question**:
>
> - With BERT-preprocessing, we mean “BERT dataset generation process”. We changed the paper accordingly and are open for change proposals. What we mean is that there is an initial process as part of the BERT algorithm, that takes a dataset (Wikipedia) and generates new sequences out of it to obtain a new dataset. Diving deep into the BERT code, we realized that it has some properties that increase the number of short sequences.

---

### Official Review · Reviewer_cYew · 2022-07-10

**Rating:** 4
**Confidence:** 4
**Soundness:** 3 good
**Presentation:** 4 excellent
**Contribution:** 2 fair

**Summary:**

This paper proposes two methods for reducing the padding tokens when packing multiple sentences into a single sequence: shortest-pack-first histogram-packing and non-negative least squares histogram-packing. Both these packing methods alter the input sequences provided to NLP models. As a result, the paper proposes basic alterations for BERT models so they can interpret packed input sequences correctly. With these changes, the paper demonstrates that a nearly 2x acceleration can be achieved versus a model using a naive token padding approach without any effect on performance.

**Questions:**

- The paper mentions that “The best speed-up is obtained with NNLSHP at depth 3 which required 28.4s for processing and ran out of memory for larger depth.”  Is there any data to support this claim? Are there circumstances where the approach would run out of memory when a depth of 3 is used?
- On lines 288-290, the paper mentions that when not using an adjusted positional embedding calculation with one of its packing algorithms, “the accuracy stalls at 71.8% and does not reach the target accuracy of 72.1%.  So overall, both adjustments are crucial.”  However, when examining the percentages and Figure 4, this claim does not seem valid.  The need for the adjustment makes sense, but is there another reason why it is a crucial change? Additionally, in Figure 4, is there a reason the lines for “packed BERT baseline” and “no pos. emb. Adjustment” end before the line for “no mask adjustment” ends?
- It feels as though an important metric is missing; for instance, why do “Faster Transformer [21], lingvo [28] fairseq [22], and RoBERTa” on line 40 all use sorted batching (SORT) to reduce padding tokens when it appears to be unviable given the results in Table 1?


**Strengths And Weaknesses:**

Strengths:
+ Potential broader applicability to many NLP tasks. While the paper mainly focuses the packing algorithms on BERT, it could also apply to nearly all transformer-related architectures.
+ Multiple factors, such as packing efficiency, processing speed-up, load balancing, convergence behavior, are considered when analyzing the validity of packing algorithms.
+ Clear writing and explanation; detailed background; helpful supplemental materials

Weaknesses:
- Incremental contribution. Bin packing is mature topic. While the specific context and constraints are different in the scope of this paper, it is unclear why many other existing bin packing algorithms cannot be applied (with reasonable modifications). The proposed two schemes are also very simple.
- Results for SORT are a bit surprising and conflicting. Section 4 concludes that SORT “cause a major overhead that exceeds the benefits of avoiding the padding”, yet Section 1 mentions that sorted batching is being used in the “Faster Transformer , lingvo, fairseq, and RoBERTa”. The seems to imply that either the evaluation results are inaccurate,or SORT has other benefits that are not discussed explicitly/clearly in the paper.
- Lack of supporting evidence for certain claims and not evaluating all parts of proposition thoroughly. For instance, on line 236, there is no provided data to support the claim that “the best speed-up is obtained with NNLSHP at depth 3”. Additionally, nearly no specifications for the training setup are provided, such as the specific GPU used. This makes the claim that the NNLSHP packing algorithm runs “out of memory for larger depth” on line 237 challenging to interpret. Also, a minor complaint is that Table 1 uses the notation >>100 and << 1.000; it would be more useful if actual numbers are provided (this is not a figure where very large/small data may not be represented well).

---

> ### Author Response · Authors · 2022-08-01
> **Response to Reviewer cYew (1/2)**
>
> We thank the reviewer for carefully reading our manuscript and for bringing up several insightful comments which we feel have helped us improve its quality. We were also encouraged by the Reviewer’s comments: “Potential broader applicability to many NLP tasks”, “it could also apply to nearly all transformer-related architectures” and “clear writing and explanation”.
>
> Let us firstly **clarify the weaknesses** addressed by the reviewer:
>
> - Incremental contribution: We can see the confusion here. This paper is not about bin-packing but addressing the multiple aspects of packing in NLP that have been neglected so far, such as data distributions, data contamination, evaluation, and documentation. We agree that the bin-packing problem has been extensively studied. However, what we prove in our manuscript is that such algorithms can be tailored to the NLP setting. We do it by proposing two new variations of the classical algorithms that we prove to 1) leave almost no padding and 2) can tackle arbitrarily large datasets in a matter of seconds. As indicated in the Appendix, more algorithms can be tailored to the NLP setting. Up to our knowledge, our manuscript is the first work to leverage classical bin-packing algorithms in the NLP context. Essentially, we did not explore more complex algorithms, because the provided ones were sufficient four our applications.
>
> - Results for SORT: We clarified this in the revised manuscript at multiple locations. The benefits of SORT are fully limited to the GPU because they require constant adjustments of the computational graph/kernels. Our reported overhead relates specifically to the Graphcore IPU, but applies also to other hardware like the Intel/Habana Gaudi accelerator or the Sambanova hardware. Even on TPUs, the changing shapes would suppress the use of the standard XLA backend, which runs several optimizations. The benefit of our proposed packing methodology is precisely that it is hardware-agnostic. To add more, even NVIDIA has recently started using it in their MLPerf submission (https://developer.nvidia.com/blog/boosting-mlperf-training-performance-with-full-stack-optimization/) in conjunction with their GPU-specific optimizations because it is faster than any other approach.
>
> - Lack of supporting evidence: We have added more information about this. The packing problem is solved on an Intel(R) Xeon(R) Gold 6138 CPU @ 2.00GHz with 80 nodes and 472G RAM, whereas BERT is trained in a Graphcore’s IPU-M2000 with 16 accelerator chips, as indicated in line 221. The >> and << symbols in Table 1 for SORT just indicate that it is a method only applicable to GPUs, and consequently trying to run it in many other accelerators is unfeasible. We could remove the SORT entry or the respective unspecific numbers from the table, if preferred. The statement about the best speed-up is supported by the bold entry in Table 1 in the last column. If more experiments are desired, we can add them to the table. Also, the respective scripts are provided with the paper, and the reader can feel free to explore more parameters. We reduced the results to the minimum required for our follow-up decisions and understanding of strength and weaknesses of the packing algorithms.

---

> > ### Author Response · Authors · 2022-08-01
> > **Response to Reviewer cYew (2/2)**
> >
> > Let us now **answer the questions**:
> >
> > - Speed-up with NNLSHP: The evidence is provided in Table 1 where the best speed-up of 1.913 is reached at a depth of 3 for NNLSHP. The provided time is another data point recorded during our experiments. If desired, we can repeat the experiment multiple times, measure variance, and observe memory occupation and add this to the appendix. However, the differences to SPFHP are so dramatic, that we did not consider this so far. Also, Section G in the appendix provides a complexity analysis of the algorithms which explains the memory explosion and in case memory would not be an issue the expected increased computational time. The complexity analysis derives formulas that show that the memory depends on the sequence length (cubed complexity). Hence, doubling the sequence length, will increase the required memory by a factor of 8. Thus, a sufficiently large sequence length, like for example 4096, would cause NNLSHP to run out of memory, whereas our SPFHP is not affected by it.
> >
> > - Clarification of ablation study: The misunderstanding here comes probably from the fact that compared to the overall curves, the difference in the final accuracies is minimal and might require to zoom in. The provided comment is supported by the experiments, we ran. In Figure 4, we provide a grey reference line at the percentage of 72.1. It is crossed by our packed approach (orange line). However, the green line where we do not adjust the positional embedding does not cross the grey line. The comment on it being in general crucial is not supported by our experiments since for the positional encoding, the difference might be considered as not significant. We correct this, by now saying: “So overall, both adjustments are crucial to avoid a reduction in performance.” We can additionally support this claim by ongoing work. We will release a new implementation that tests our packing implementation and verifies that loss and gradient match exactly if and only if all our proposed adjustments are applied. If the attention is not filtered, tokens from completely different sequences contaminate and add noise that drastically reduces performance. The impact on the positional encoding is less severe, since the impact of positional encoding on performance is less, especially compared to the attention masking. Furthermore, the positional encoding is containing still some correct information. Mostly, for the second and third sequence in the pack, we can interpret the lack of adjustment of the positional encoding that the model does not know, where the sequence begins and can only benefit from some relative, but not absolute distance.
> >
> >   When we ran the experiments without the attention adjustment, we were wondering if convergence is just slower but still can reach the target after longer training. We can see that this can be confusing. If the reviewer agrees, we will remove the extended run information.
> > - Missing metric to support SORT: The missing detail in the table is not a metric but the device that is used to measure the overhead (IPU as mentioned in line 221) as mentioned in “sorted batching (SORT) is highly efficient in avoiding padding but the resulting different computational graphs cause a major overhead on the IPU that exceeds the benefits of avoiding the padding”. This is addressed now with an extended experiment description and addition to the caption and table description. SORT is limited to GPUs and even there, it can limit performance. In the introduction, we mention “these approaches require substantial hardware-specific low-level code optimizations as discussed more in detail in Sections ...”. We also provide a reference where our approach was used in TensorFlow using the Intel/Habana Gaudi accelerator with the same performance gain. We will add references to a new PyTorch Huggingface implementation and a more recent NVIDIA implementation in the final paper version.

---

> ### Comment · Reviewer_cYew · 2022-08-09
> **Acknowledgment of Author Response**
>
> I would like to thank the authors for responding to my questions and concerns. I have reviewed the response and will discuss with fellow reviewers in the next stage.

---

### Official Review · Reviewer_51Ua · 2022-07-10

**Rating:** 4
**Confidence:** 3
**Soundness:** 3 good
**Presentation:** 2 fair
**Contribution:** 2 fair

**Summary:**

The paper proposes speeding large-scale pre-training by constructing the data pipeline with the explicit goal of minimizing the need for padding tokens, which only lead to wasted computation. The approach is to consider the dataset as a whole and apply bin-packing algorithms to merge several short examples into one longer example, which is done up-front as part of preprocessing. Two bin packing strategies (SPFHP and NNLSHP) are proposed and evaluated. Pre-training may be sped up by up to 2x compared to a naive approach.

**Questions:**

none

**Limitations:**

Yes

**Strengths And Weaknesses:**

The main strengths include, of course, the ability to achieve sizable speed-ups in practice. Moreover, the idea of having a whole-dataset preprocessing step aimed at optimizing it for accelerator use is a very general one. Bringing this to the attention of the community, and suggesting how to think about this in terms of concepts/terminology/abstractions, could lead to less computationally wasteful practices across the board in the field. It's especially helpful to see a quantitative analysis that reveals that padding is a big source of inefficiency in pre-training, where one might naively assume that it's not an issue because the underlying Wikipedia articles are actually quite long.

In terms of weaknesses, my immediate impression reading the paper for the first time is that many aspects of the idea of packing examples together are not, in fact, novel. The paper acknowledges that frameworks already concatenate short examples as a way to speed up training, and that this practice is commonly referred to as "packing". The claim in Sec 3.2.2 that the paper "propose[s] directly masking the attention matrix" is not novel either. (As prior art, see [tensor2tensor](https://github.com/tensorflow/tensor2tensor/commit/c9144dfa5f514cab529f487b069415daee5e211e#diff-3c271923bb62bdd35f3b0f6a2c94ea320825d834bbf51334a9acbc04fbea9763R538) for code essentially the same as Figure 2). Likewise for positional embedding and loss adjustment.

It seems that the actual novelty of the paper is in highlighting packing efficiency as a key criterion in designing a data pipeline, proposing to optimize this criterion at a whole-dataset level (instead of ad-hoc or streaming-based approaches), as well as showing that this can be done in a way that is scalable to large-scale datasets. At least for me, the writing of the paper did not succeed in delineating the scope of the contribution from what has been done in prior work. Additionally, the term "packing" has acquired a certain meaning in the context of modern transformer training, and I would recommend that the authors adopt distinct terminology to differentiate their approach. [Edit: to clarify, I am not referring to the use of the term "packing" in the title (which is fine), but usages like "packed" in Table 3 or even "packed (our approach)" in Figure 5. For example, looking at Table 3 / Figure 5 in isolation would be confusing to readers that have a different prior conception of what the term "packing" refers to].

I also would have found it helpful if the paper had made explicit its assumption that packing takes place during an up-front pre-processing stage where the whole dataset is available for random access, and not in an on-line streaming manner. Approaches like padding and greedy concatenation would all work when streaming the data from storage one example at a time, while the proposed method does not. Even if engineering practice in the narrow area of pre-training does data processing up-front, the corresponding academic literature often doesn't necessarily go into this and sometimes presents a picture that leaves open an interpretation where padding/packing/batching or even tokenization are done via streaming processing.

In summary, the main weakness of the paper is how it frames the discussion for an academic audience, including how it contrasts its contributions from the literature as a whole.

---

> ### Author Response · Authors · 2022-08-01
> **Response to Reviewer 51Ua**
>
> We were very encouraged by the Reviewer's comments: “Bringing this to the attention of the community […] could lead to less computationally wasteful practices" and “it's especially helpful to see a quantitative analysis that reveals that padding is a big source of inefficiency”. The reviewer claims that the main reason for rejection of the paper is “how it frames the discussion for an academic audience, including how it contrasts its contributions from the literature as a whole”. There are multiple ways of framing the presentation of a contribution and we have already incorporated the feedback from the reviewer into the manuscript. We are receptive to hear about more suggestions to make the presentation of our contributions clearer.
>
> We added the link to the Github diff of the deprecated Google tensor2tensor library to our related work and to our masking section. More references could be added if provided. Nevertheless, the tensor2tensor implementation does not provide sufficient documentation or any evaluation to be considered published work or being useful for the research community. To make it useful, it requires the data analysis, descriptions, and comparisons of concepts that we provide in our paper. On the contrary, our aim was precisely to put forward a packing algorithm that is supported by a peer-reviewed paper with open-source code and convergence results on a real dataset. Thanks to this effort, the hardware manufacturers NVIDIA, Graphcore and Intel have recently established our approach as the standard for performance benchmarking in NLP. This shows that previously there was no awareness of the necessity and power of our approach and the potential implementation in tensor2tensor.
>
> We did not explore online approaches in our paper, however several of our approaches and evaluations can be transferred to the online scenario, too. Efficiency will slightly reduce but with the right sample buffer getting from 50% to 90% efficiency should be easily possible. We state now more explicitly that we look at the whole dataset and offline packing algorithms. In the previous version, this was also mentioned in between the lines and in the Appendix section on bin-packing.
>
> To the best of our knowledge, packing is just referring to putting things together in a finite space. In the context of NLP, this means to concatenate sequences, which can come with many facets and forms. We are not aware of any other “certain meaning in the context of modern transformer training”. Given, that there is no paper that focuses on packing in NLP, we believe the name is well chosen. Additionally, paper titles are supposed to be short and ours is already quite long to make clear which topic we are addressing. However, we are open to naming proposals for our approaches and the title of the paper to avoid confusion. Would it help to use “Efficient Sequence Packing **and Processing** without Cross-contamination: Accelerating Large Language Models without Impacting Performance” instead as a title?
>
> To conclude, we are grateful that the reviewer has brought up several insightful suggestions which we feel have helped us improve the quality of the manuscript. We are receptive to hear about more feedback and are looking forward to engaging more with the reviewer about further changes to improve how we frame the discussion.

---

### Author Response · Authors · 2022-08-01
**General response to initial reviews**

These reviews make great suggestions for clarification and improvement of this paper.  We believe we have answered the various technical questions.  We are encouraged that the reviewers generally agree that packing, while looking like an engineering detail, in fact touches interesting research questions.  As mentioned in some of our answers, this is a methodology that is exemplified for the most part in code bases, with little analysis, ablation, or theoretical underpinnings.  We believe that aggregating this diffused knowledge in a coherent framework is a benefit to the community, and this is on top of the novelty of our specific approach, which is already acknowledged by its adoption by other teams in the community.

---

### Meta-Review · Area_Chair_UPif · 2022-08-29

**Recommendation:** Reject
**Confidence:** Certain

**Metareview:**

This paper draws attention to the importance of good packing to avoid padding when creating batches. This problem is indeed important in practice and the paper does a good job studying the paper. That being said, the machine learning novelty seems limited. One reviewer was strongly supportive of acceptance while the other two thought this paper was below the cut-off. The meta reviewer thinks that there isn't sufficient ML novelty for NeurIPS.

**Award:**

No

---

### Decision · Program_Chairs · 2022-09-14

Reject